# Boosting Adversarial Transferability of Vision-Language Pre-trained Models via Optimal Transport

## Abstract

Vision-language pre-trained models (VLP) exhibit remarkable capabilities in processing images and textual information. However, they are vulnerable to multimodal adversarial examples. Notably, adversarial examples generated for a specific model can potentially deceive different models, known as adversarial transferability. The potential threats posed by adversarial transferability to models in practical applications have heightened interest in studying the transferability of adversarial examples. Recent works have indicated that leveraging data augmentation and image-text modal interactions can significantly enhance the transferability of adversarial examples for VLP models. A crucial aspect of this improvement hinges on how information between different modalities is aligned. Despite this, they have overlooked the critical issue of finding the optimal alignment between data-augmented image-text pairs. This oversight creates adversarial examples overly customized to the source model, consequently restricting their transferability potential. In our research, we first explore the interplay between image sets produced through data augmentation and their corresponding text sets. We find that augmented image samples can align more effectively with specific texts while exhibiting less relevance to others. Motivated by this, we propose an Optimal Transport-based Adversarial Attack, *dubbed* OT-Attack. The proposed method formulates the features of image and text sets as two distinct distributions and employs optimal transport theory to determine the most efficient mapping between them. This optimal mapping informs our generation of adversarial examples to enhance their transferability. Extensive experiments across various network architectures and datasets in image-text matching tasks show that our OT-Attack is more transferable to unseen target models than existing methods.

## 1 Introduction

Vision-Language Pre-trained (VLP) models have shown outstanding performance in various downstream tasks, including image-text matching (Cao et al., 2022; Li et al., 2023), image captioning (Hossain et al., 2019; Ghandi et al., 2023), visual question answering (Li et al., 2022), and visual grounding (Deng et al., 2018; Yang et al., 2023). Despite their impressive capabilities, these models encounter significant security challenges in real-world applications (Lei et al., 2021; Zhou et al., 2020; Bao et al., 2022; Hu et al., 2022).

Existing works have demonstrated that adversarial examples perturbed on white-box models remain effective on certain black-box models (Goodfellow et al., 2014; Papernot et al., 2016). It indicates that adversarial examples generated via a proxy model can still mislead the prediction of black-box models due to their transferability (Xie et al., 2019; Lin et al., 2019; Dong et al., 2019; Jia et al., 2020; Long et al., 2022; Jia et al., 2022). A ideal attack scenario, in reality, is one where adversarial examples remain effective even in the absence of detailed knowledge about the model's inner workings, such as its model architecture, weights, and gradients, etc (Han et al., 2023; Gu et al., 2023). Motivated by the practical significance of transfer-based adversarial attacks and adversarial transferability (Gubri et al., 2022; Qin et al., 2022; Byun et al., 2022; Waseda et al., 2023), in this paper, we primarily study the transferability of adversarial examples across VLP models.

Zhang et al. (2022) proposed Co-Attack, which combines modalities using image-text pairs to improve transferability. Further, Lu et al. (2023) developed the Set-level Guidance Attack (SGA),

advancing Co-Attack by employing data augmentation and multiple textual descriptions for set-level alignment and intermodal guidance, achieving SOTA results in VLP models. However, as illustrated in Figure 1, different captions of the same image may focus on different contents. The critical limitation of SGA lies in its approach of averaging the alignment between sets of captions and images without considering the crucial matches between specific captions and corresponding image contents. This generalized matching strategy fails to ensure optimal alignment, especially after images have undergone data augmentation processes such as zooming, which can lead to significant misalignments with their captions. Con-

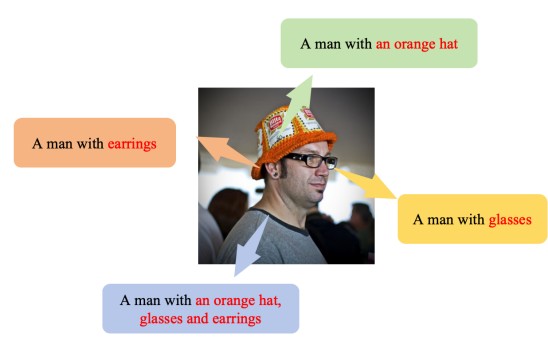

Figure 1: An image from the Flickr30K often has captions that focus on different parts of the image, meaning one caption may be highly relevant to a specific region but less so to others.

sequently, this approach may reduce the efficacy of data augmentation and modality interactions for improving adversarial transferability.

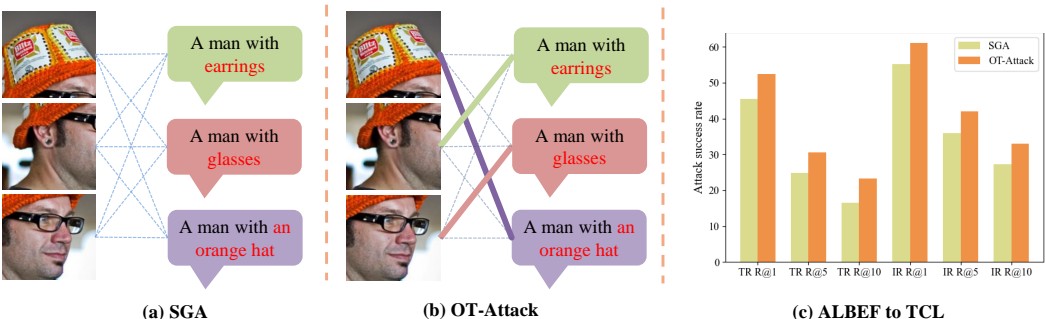

**(a) SGA**  **(b) OT-Attack**  **(c) ALBEF to TCL**

Figure 2: Comparative analysis of Set-level Guidance Attack (SGA) methods and their ITR attack success rates. Panel **(a)** illustrates the conventional SGA approach where image and text sets are averaged to establish pair-wise matches. Panel **(b)** showcases our proposed method, OT-Attack, where images are matched to texts based on optimal transport theory to enhance matching accuracy. Panels **(c)** depict the attack success rates for our method OT-Attack versus traditional SGA, with ALBEF serving as the source and TCL serving as the target. The bar charts show that our adversarial examples outperform SGA in all metrics, effectively disrupting ITR performance.

In this paper, we address this issue by incorporating the theory of optimal transport (Villani et al., 2009). We treat the feature sets of augmented images and texts as two distinct distributions and aim to establish the optimal transport scheme between them. The distinction between our method and SGA, along with a comparative overview of the results, is depicted in Figure 2. In detail, we integrate optimal transport theory to analyze data-augmented image sets and text sets as distinct distributions. This holistic consideration allows us to incorporate similarity into the cost matrix and calculate the optimal transport scheme. Consequently, we compute the total transfer cost between these distributions, guiding the generation of adversarial examples. Our method achieves a more balanced matching relationship between the augmented image and text sets, leading to more effective alignment and improving the transferability of adversarial examples. Experiments conducted on various models including ALBEF (Li et al., 2021), TCL (Yang et al., 2022), and CLIP (Radford et al., 2021), and utilizing well-known datasets like Flickr30K (Plummer et al., 2015) and MSCOCO (Lin et al., 2014), quantitatively demonstrate the effectiveness of our approach.

The key contributions of this paper are summarized in three aspects:

1. We propose a framework that improves the SGA by ensuring a balanced match between image and text sets after data augmentation.

2. We innovatively utilize Optimal Transport theory in examining adversarial example transferability in VLP models, promoting a more profound and thorough alignment between data-augmented images and textual descriptions.

3. Extensive experiments establish that our method generates adversarial examples with superior transferability compared to existing state-of-the-art techniques. Furthermore, our OT-Attack can successfully break current business models like GPT-4 and Bing Chat.

## 2 BACKGROUND AND RELATED WORK

**Vision-Language Pre-trained Models.** Vision-language pre-training (VLP) (Chen et al., 2023) is a pivotal technique in augmenting multimodal task performance, capitalizing on extensive pre-training with image-to-text pairs. Traditionally, much of the research in this area has relied on pre-trained object detectors, using region features to create vision-language representations. However, the advent of Vision Transformer (ViT) (Dosovitskiy et al., 2020; Han et al., 2022) has instigated a methodological shift. Increasingly, studies advocate adopting ViT in image encoding, which involves an end-to-end process of transforming inputs into patches. VLP models can be broadly classified into fused and aligned VLP models. Fused VLP models, as exemplified by architectures like ALBEF (Li et al., 2021) and TCL (Yang et al., 2022), utilize individual unimodal encoders for processing token and visual feature embeddings. These models then employ a multimodal encoder to amalgamate image and text embeddings, crafting comprehensive multimodal representations. Conversely, aligned VLP models use unimodal encoders to process image and text modality embeddings independently.

**Vision-Language Tasks.** *Image-text retrieval.* Image-Text Retrieval (ITR) (Cao et al., 2022; Li et al., 2023) is a task that retrieves relevant instances from a database using one modality (image or text) to query the other. It splits into image-to-text retrieval (TR) and text-to-image retrieval (IR). Models like ALBEF and TCL calculate semantic similarity scores between image-text pairs for initial ranking, then employ a multimodal encoder for final ranking. Conversely, models like CLIP (Radford et al., 2021) directly rank based on similarity in an unimodal embedding space, showcasing varied ITR methodologies. *Image captioning.* Image captioning (Hossain et al., 2019; Ghandi et al., 2023) generates textual captions for images and is crucial in VLP models. This task requires converting visual content into coherent, contextually relevant text, which differs from image-text retrieval. *Visual grounding.* Visual Grounding (Deng et al., 2018; Yang et al., 2023) entails identifying and locating objects or regions in an image per language descriptions, requiring precise text mapping to visual elements.

**Transferability of Adversarial Examples.** Co-Attack by Zhang et al. (2022) integrates visual and textual attacks, exploiting VLP model multimodality. The Set-level Guidance Attack (SGA) advances this by aligning augmented images with multiple texts, enhancing adversarial example transferability across black-box models. The shift from individual to integrated attacks like Co-Attack and SGA illustrates the evolution of adversarial strategy against VLP models.

**Optimal Transport.** Optimal Transport (OT), a concept first introduced by Monge (Villani et al., 2009), its unique ability to match distributions has led to its widespread application in various theoretical and practical areas. This includes its use in generative models and structural alignments involving sequences (Arjovsky et al., 2017), graphs (Xu et al., 2019), and image matching (Zhang et al., 2020; Liu et al., 2021; Zhao et al., 2021).

## 3 APPROACH

### 3.1 THREAT MODEL

We analyze two scenarios: **white-box** and **black-box** attacks. In a **white-box model** $M_{\text{white}}$, the adversary has complete access to the model's architecture, parameters, and gradients, which allows for direct optimization to generate adversarial examples. In contrast, a **black-box model** $M_{\text{black}}$ is opaque, restricting the adversary to indirect methods based on observed outputs or behavior.

This work focuses on generating adversarial examples on a white-box model and leveraging these examples to attack a black-box model. This approach is aimed at evaluating the transferability of adversarial examples and the effectiveness of attack strategies. A well-designed loss function $\mathcal{L}$

in the white-box setting plays a critical role in enhancing attack success rates. In this regard, the proposed method explores the integration of optimal transport loss into adversarial attacks, which will be discussed in detail in the following sections.

For $M_{\text{white}}$, the adversary seeks to maximize a loss function $\mathcal{L}$ under a constraint on the magnitude of the perturbation:

$$\Delta^* = \arg \max_{\Delta} \mathcal{L}(f_{\text{white}}(I_{\text{orig}})), \quad \text{subject to} \quad ||\Delta||_p \leq \epsilon, \tag{1}$$

where $|| \cdot ||_p$ denotes the $p$-norm, and $\epsilon$ defines the permissible visual deviation from the original image. The resulting adversarial example is computed as:

$$I_{\text{adv}} = I_{\text{orig}} + \Delta^*. \tag{2}$$

## 3.2 THE PROPOSED METHOD

### 3.2.1 SYMBOL CONVENTIONS

In this section, we describe the sources of image and text features utilized in our framework, followed by a discussion on how traditional attack methods define the loss function $\mathcal{L}$.

Given an original set of images $\mathcal{I}$ and a set of image enhancement factors $\mathcal{A}$, we construct the augmented image set $\mathcal{I}_{aug}$ by applying an image enhancement method $f_{\text{enhance}}$ to each image $I \in \mathcal{I}$ for every factor $\alpha \in \mathcal{A}$:

$$\mathcal{I}_{aug} = \bigcup_{\alpha \in \mathcal{A}} \left( f_{\text{enhance}}(I, \alpha) \right),$$

where $f_{\text{enhance}}$ represents a generic image enhancement operation. Using the augmented image set $\mathcal{I}_{aug}$ and an original text set $\mathcal{T}$, we extract their feature representations via encoders. Specifically, the image encoder $\phi$ and text encoder $\varphi$ produce $\mathbf{F}_{img} = \phi(\mathcal{I}_{aug})$ and $\mathbf{X}_{txt} = \varphi(\mathcal{T})$, where $\mathbf{F}_{img}$ and $\mathbf{X}_{txt}$ are the image and text feature representations, respectively.

The similarity matrix $\mathbf{S}$, representing the pairwise similarity between image and text features, is computed as:

$$\mathbf{S} = \mathbf{F}_{img} \odot \mathbf{X}_{txt},$$

where $\odot$ denotes matrix multiplication.

Traditional attack methods often define the loss function $\mathcal{L}$ using the similarity matrix $\mathbf{S}$. A commonly employed formulation is as follows:

$$loss_{ori} = - \left( \sum_i \mathbf{S}_i \right)_{\text{mean}}, \tag{3}$$

where the summation $\sum_i \mathbf{S}_i$ is taken over the last dimension of the similarity matrix $\mathbf{S}$. The mean of this summation is then computed to obtain the final loss value $loss_{ori}$. This formulation encourages the adversarial examples to maximize dissimilarity between the features of augmented images and original texts, facilitating the generation of effective attacks on the white-box model.

However, a significant limitation often hinders traditional attack methods: the generated adversarial examples tend to overfit the source (white-box) model. During optimization, the perturbations are excessively tailored to exploit the white-box model's specific features and decision boundaries. While this overfitting improves attack success on the source model, it severely reduces the transferability of adversarial examples to black-box models. This lack of transferability is a critical challenge, as it undermines the effectiveness of adversarial attacks in practical scenarios. Experimental results in the Appendix H substantiate this observation, highlighting the need for methods to balance attack success on the source model while enhancing generalization to unseen models.

### 3.2.2 OPTIMAL TRANSPORT

**Defining Source (P) and Target (Y) Distributions.** In the Optimal Transport framework, we begin by defining two fundamental distributions: the source distribution $\mathbf{P}$ and the target distribution $\mathbf{Y}$. These distributions represent the starting and ending points of the transportation process in the

Optimal Transport problem. Specifically, the source distribution $\mathbf{P} = (p_1, p_2, \ldots, p_n)$ and the target distribution $\mathbf{Y} = (y_1, y_2, \ldots, y_m)$ describe the quantities to be transported from and to each respective location.

**The Transportation Matrix T.** In the context of Optimal Transport, the matrix $\mathbf{T} = [T_{ij}]$ of size $n \times m$ is referred to as the transportation matrix. Each element $T_{ij}$ represents the amount of a commodity or resource transported from the $i$-th source in $\mathbf{P}$ to the $j$-th target in $\mathbf{Y}$. This matrix effectively captures the transportation scheme between the sources and targets.

The matrix $\mathbf{T}$ must satisfy certain constraints to ensure an optimal transportation plan. The Marginal Constraints are:

$$\sum_{j=1}^{m} T_{ij} = p_i, \ \forall i \in \{1, \ldots, n\}, \quad \text{and} \quad \sum_{i=1}^{n} T_{ij} = y_j, \ \forall j \in \{1, \ldots, m\}. \tag{4}$$

These constraints ensure that the total transported amount from each source $i$ and to each target $j$ equals the respective supply $p_i$ and demand $y_j$.

Additionally, the Non-Negativity Constraint is imposed:

$$T_{ij} \geq 0, \ \forall i \in \{1, \ldots, n\}, \forall j \in \{1, \ldots, m\}. \tag{5}$$

This condition ensures that all transport amounts $T_{ij}$ are non-negative, reflecting the practical impossibility of negative transportation.

**Modeling the Optimal Transport Problem.** With the aforementioned definitions and constraints established, the Optimal Transport (OT) problem can be formulated as follows:

$$OT(\mathbf{P}, \mathbf{Y}, \mathbf{C}) = \min_{\mathbf{T} \in \Pi(\mathbf{r}, \mathbf{c})} \sum_{i,j} T_{ij} C_{ij}, \tag{6}$$

where $\mathbf{C}$ denotes the cost matrix, with each element $C_{ij}$ representing the cost of transporting a unit from source $p_i$ to target $y_j$. The matrix $\mathbf{T}$ represents the transportation plan, and $\Pi(\mathbf{r}, \mathbf{c})$ defines the set of all feasible transportation plans that satisfy the marginal constraints.

To address computational challenges in high-dimensional spaces, the Sinkhorn distance is widely used in OT due to its efficiency and scalability. Traditional OT approaches, based on linear programming, often struggle with computational intensity and poor scalability as data dimensionality increases. In contrast, the Sinkhorn distance introduces entropy regularization into the OT formulation, significantly improving tractability and enabling gradient-based optimization.

This regularization is controlled by a parameter $\lambda$, which balances the trade-off between accuracy and computational efficiency. Larger $\lambda$ values yield results closer to traditional OT at the cost of higher computational expense, while smaller $\lambda$ values accelerate computations at the expense of slight bias. The Sinkhorn Optimization Process is:

$$OT_\lambda(\mathbf{P}, \mathbf{Y}, \mathbf{C}) = \min_{\mathbf{T} \in \Pi(\mathbf{r}, \mathbf{c})} \sum_{i,j} T_{ij} C_{ij} + \lambda H(\mathbf{T}) \tag{7}$$

The algorithm of the proposed OT-Attack is summarized in Algorithm 1 of Appendix A.

### 3.2.3 Calculating Loss through Optimal Transport

The Optimal Transport loss $\text{loss}_{OT}$ is computed using the feature representations of augmented images $\mathbf{F}_{img}$, original texts $\mathbf{X}_{txt}$, and the similarity matrix $\mathbf{S}$.

First, the cost matrix $\mathbf{C}$ is defined as $\mathbf{C} = 1 - \mathbf{S}$, transforming similarity scores into a cost structure. Next, the exponentiated negative cost matrix $\mathbf{K}$ is computed for the Sinkhorn iterations, given by $\mathbf{K} = \exp\left(-\frac{\mathbf{C}}{\lambda}\right)$, where $\lambda$ is a small positive regularization parameter. The Optimal Transport loss is then calculated as:

$$\text{loss}_{OT} = \sum_{i,j} T_{ij} C_{ij} \tag{8}$$

where $T_{ij}$ in $\mathbf{T}$ represents the optimal 'transport' of features from the $i$-th element in $\mathbf{F}_{img}$ to the $j$-th element in $\mathbf{X}_{txt}$, and $C_{ij}$ is the corresponding cost in $\mathbf{C}$.

Table 1: Attack success rate at Rank 1 (ASR @ R1) of different adversarial attack methods for text-image retrieval (IR) and text-image retrieval (TR) tasks using the Flickr30K dataset.

| Source | Attack | ALBEF | | TCL | | CLIP$_{ViT}$ | | CLIP$_{CNN}$ | |
|---|---|---|---|---|---|---|---|---|---|
| | | TR R@1 | IR R@1 | TR R@1 | IR R@1 | TR R@1 | IR R@1 | TR R@1 | IR R@1 |
| **ALBEF** | PGD | 52.45 | 58.65 | 3.06 | 6.79 | 8.69 | 13.21 | 10.34 | 14.65 |
| | BERT-Attack | 11.57 | 27.46 | 12.64 | 28.07 | 29.33 | 43.17 | 32.69 | 46.11 |
| | Sep-Attack | 65.69 | 73.95 | 17.60 | 32.95 | 31.17 | 45.23 | 32.82 | 45.49 |
| | Co-Attack | 77.16 | 83.86 | 15.21 | 29.49 | 23.60 | 36.48 | 25.12 | 38.89 |
| | SGA | 97.24 | 97.28 | 45.42 | 55.25 | 33.38 | 44.16 | 34.93 | 46.57 |
| | OT-Attack (Ours) | 95.93 | 95.86 | **52.37** | **61.05** | **34.85** | **47.10** | **42.33** | **53.03** |
| **TCL** | PGD | 6.15 | 10.78 | 77.87 | 79.48 | 7.48 | 13.72 | 10.34 | 15.33 |
| | BERT-Attack | 11.89 | 26.82 | 14.54 | 29.17 | 29.69 | 44.49 | 33.46 | 46.07 |
| | Sep-Attack | 20.13 | 36.48 | 84.72 | 86.07 | 31.29 | 44.65 | 33.33 | 45.80 |
| | Co-Attack | 23.15 | 40.04 | 77.94 | 85.59 | 27.85 | 41.19 | 30.74 | 44.11 |
| | SGA | 48.91 | 60.34 | 98.37 | 98.81 | 33.87 | 44.88 | 37.74 | 48.30 |
| | OT-Attack (Ours) | **57.32** | **65.83** | 97.81 | 98.01 | **34.72** | **47.16** | **43.44** | **54.12** |
| **CLIP$_{ViT}$** | PGD | 2.50 | 4.93 | 4.85 | 8.17 | 70.92 | 78.61 | 5.36 | 8.44 |
| | BERT-Attack | 9.59 | 22.64 | 11.80 | 25.07 | 28.34 | 39.08 | 30.40 | 37.43 |
| | Sep-Attack | 9.59 | 23.25 | 11.38 | 25.60 | 79.75 | 86.79 | 30.78 | 39.76 |
| | Co-Attack | 10.57 | 24.33 | 11.94 | 26.69 | 93.25 | 95.86 | 32.52 | 41.82 |
| | SGA | 13.40 | 27.22 | 16.23 | 30.76 | 99.08 | 98.94 | 38.76 | 47.79 |
| | OT-Attack (Ours) | **14.29** | **29.28** | **16.58** | **33.49** | 98.65 | 98.52 | **43.55** | **50.50** |
| **CLIP$_{CNN}$** | PGD | 2.09 | 4.82 | 4.00 | 7.81 | 1.10 | 6.60 | 86.46 | 92.25 |
| | BERT-Attack | 8.86 | 23.27 | 12.33 | 25.48 | 27.12 | 37.44 | 30.40 | 40.10 |
| | Sep-Attack | 8.55 | 23.41 | 12.64 | 26.12 | 28.34 | 39.43 | 91.44 | 95.44 |
| | Co-Attack | 8.79 | 23.74 | 13.10 | 26.02 | 28.79 | 40.03 | 94.76 | 96.89 |
| | SGA | 11.42 | 24.80 | **14.91** | 28.82 | 31.24 | 42.12 | 99.24 | 99.49 |
| | OT-Attack (Ours) | **11.57** | **26.24** | **14.91** | **30.52** | **35.63** | **48.20** | 99.39 | 99.32 |

This formulation of loss$_{OT}$ captures the minimal cost required to align the feature representations of augmented images with those of the original texts. By leveraging the overall feature distribution, it facilitates the generation of more effective adversarial examples. Importantly, this method addresses potential overfitting issues inherent in relying solely on a similarity matrix as the loss metric. Details of the process are provided in Algorithm 2 in Appendix B.

## 4 EXPERIMENTS

### 4.1 SETTINGS

**VLP Models.** To evaluate adversarial examples' transferability and our framework's performance, we examined two Vision-Language Pre-trained (VLP) model types: fused and aligned VLPs. Fused VLPs, like ALBEF (Li et al., 2021) and TCL (Yang et al., 2022), process images and text together with shared layers, using a 12-layer ViT-B/16 (Dosovitskiy et al., 2020) for visuals and two 6-layer transformers for image and text data. Aligned VLPs, such as CLIP (Radford et al., 2021) variants (CLIP$_{ViT}$ with ViT-B/16 and CLIP$_{CNN}$ with ResNet-101 (He et al., 2016)), process data separately before aligning it in later stages. We assessed cross-task attack success on image captioning using BLIP, with adversarial examples generated using TCL.

**Datasets.** For the image-text retrieval task, our study utilized two datasets renowned for their breadth and depth: Flickr30K (Plummer et al., 2015) and MSCOCO (Lin et al., 2014). Flickr30K boasts a diverse corpus of 31,783 images, while MSCOCO expands the dataset considerably with 123,287 images. A salient characteristic shared by both is the quintuple of descriptive captions accompanying each image, providing a valuable asset for the assessment of our image-text retrieval approach. For the task of Visual Grounding, we employed the RefCOCO+ (Yu et al., 2016) dataset, which further enriched our cross-task attack effectiveness analysis.

**Baselines** In our research involving Vision-Language Pre-trained (VLP) models, we implemented several prevalent adversarial attack methods as baselines. These included using PGD (Madry et al., 2017) exclusively on images, applying BERT-Attack (Li et al., 2020) only to texts, and separately utilizing PGD and BERT-Attack on both images and texts without integrating inter-modality interactions, a technique designated as Sep-Attack. Additionally, we employed Co-Attack (Zhang et al., 2022), which integrates information between individual image-text pairs, and Set-level Guidance

Table 2: Attack success rate at Rank 1 (ASR @ R1) of different adversarial attack methods for text-image retrieval (IR) and text-image retrieval (TR) tasks using the MSCOCO dataset.

| Source | Attack | ALBEF | | TCL | | CLIP$_{ViT}$ | | CLIP$_{CNN}$ | |
|--------|--------|-------|-------|-----|-----|--------------|--------------|--------------|--------------|
| | | TR R@1 | IR R@1 | TR R@1 | IR R@1 | TR R@1 | IR R@1 | TR R@1 | IR R@1 |
| ALBEF | PGD | 76.70 | 86.30 | 12.46 | 17.77 | 13.96 | 23.1 | 17.45 | 23.54 |
| | BERT-Attack | 24.39 | 36.13 | 24.34 | 33.39 | 44.94 | 52.28 | 47.73 | 54.75 |
| | Sep-Attack | 82.6 | 89.88 | 32.83 | 42.92 | 44.03 | 54.46 | 46.96 | 55.88 |
| | Co-Attack | 79.87 | 87.83 | 32.62 | 43.09 | 44.89 | 54.75 | 47.3 | 55.64 |
| | SGA | 96.75 | 96.95 | 58.56 | 65.38 | 57.06 | 62.25 | 58.95 | 66.52 |
| | OT-Attack (Ours) | 95.41 | 95.8 | **63.44** | **68.9** | **58.79** | **65.87** | **63.56** | **72.16** |
| TCL | PGD | 10.83 | 16.52 | 59.58 | 69.53 | 14.23 | 22.28 | 17.25 | 23.12 |
| | BERT-Attack | 35.32 | 45.92 | 38.54 | 48.48 | 51.09 | 58.8 | 52.23 | 61.26 |
| | Sep-Attack | 41.71 | 52.97 | 70.32 | 78.97 | 50.74 | 60.13 | 51.9 | 61.26 |
| | Co-Attack | 46.08 | 57.09 | 85.38 | 91.39 | 51.62 | 60.46 | 52.13 | 62.49 |
| | SGA | 65.93 | 73.3 | 98.97 | 99.15 | 56.34 | 63.99 | 59.44 | 65.7 |
| | OT-Attack (Ours) | **71.64** | **78.38** | 98.69 | 98.78 | **58.64** | **65.75** | **63.45** | **72.01** |
| CLIP$_{ViT}$ | PGD | 7.24 | 10.75 | 10.19 | 13.74 | 54.79 | 66.85 | 7.32 | 11.34 |
| | BERT-Attack | 20.34 | 29.74 | 21.08 | 29.61 | 45.06 | 51.68 | 44.54 | 53.72 |
| | Sep-Attack | 23.41 | 34.61 | 25.77 | 36.84 | 68.52 | 77.94 | 43.11 | 49.76 |
| | Co-Attack | 30.28 | 42.67 | 32.84 | 44.69 | 97.98 | 98.8 | 55.08 | 62.51 |
| | SGA | 33.41 | 44.64 | 37.54 | 47.76 | 99.79 | 99.79 | 58.93 | 65.83 |
| | OT-Attack (Ours) | **35.11** | **46.48** | **38.52** | **50.32** | 99.69 | 99.75 | **62.16** | **68.96** |
| CLIP$_{CNN}$ | PGD | 7.01 | 10.62 | 10.08 | 13.65 | 4.88 | 10.7 | 76.99 | 84.2 |
| | BERT-Attack | 23.38 | 34.64 | 24.58 | 29.61 | 51.28 | 57.49 | 54.43 | 62.17 |
| | Sep-Attack | 26.53 | 39.29 | 30.26 | 41.51 | 50.44 | 57.11 | 88.72 | 92.49 |
| | Co-Attack | 29.83 | 41.97 | 32.97 | 43.72 | 53.1 | 58.9 | 96.72 | 98.56 |
| | SGA | 31.61 | 43 | 34.81 | 45.95 | 56.62 | 60.77 | 99.61 | 99.8 |
| | OT-Attack (Ours) | **32.9** | **44.03** | **36.07** | **48.17** | **61.14** | **67.79** | 99.16 | 99.59 |

Attack (SGA) (Lu et al., 2023), which utilizes guidance information across modalities between sets. Each baseline was tested under identical conditions for a consistent comparative analysis.

**Adversarial Attack Configuration.** To validate our framework's effectiveness, we followed the experimental setup outlined in the SGA for generating adversarial examples in both visual and textual domains. We generated adversarial visual examples using the Projected Gradient Descent (PGD) method (Madry et al., 2017) with specific settings: a perturbation limit of $\epsilon_v = \frac{2}{255}$, a step size of $\alpha = \frac{0.5}{255}$, and $T = 10$ iterations. For textual examples, we used BERT-Attack (Li et al., 2020) with a disturbance limit of $\epsilon_t = 1$ and a vocabulary size of $W = 10$. These settings were consistently applied in our experimentation with Sep-Attack and Co-Attack. Specifically for Co-Attack, we additionally utilized the similarity between individual image pairs as a loss metric, guiding the generation of adversarial examples through inter-modality interactions. In the case of SGA, we adhered to the experimental conditions outlined in its original publication, notably enhancing images by rescaling them to five distinct sizes $\{0.5, 0.75, 1.0, 1.25, 1.5\}$. To further demonstrate the effectiveness of our method, we employed the same experimental setup as SGA, including adopting a perturbation limit of $\epsilon_v = \frac{2}{255}$. Additionally, we integrated the Sinkhorn algorithm (Cuturi, 2013) for calculating the optimal transport plan, using a regularization parameter $\lambda = 0.1$ to balance transport cost minimization and plan smoothness. To prevent the iteration process from becoming infinite, we set a convergence threshold $thresh = 1e - 2$.

**Evaluation Criteria.** In our study, the robustness and transferability of the adversarial attacks are quantitatively assessed using the Attack Success Rate (ASR). ASR is a crucial metric that measures the proportion of successful adversarial examples out of the total number of attacks conducted. A higher ASR is indicative of increased transferability of the adversarial examples, signifying the effectiveness of the attack in compromising the model under various conditions. The ASR is computed as $ASR = \frac{N_{\text{success}}}{N_{\text{total}}} \times 100\%$ where $ASR$ denotes the Attack Success Rate, $N_{\text{success}}$ represents the number of successful attacks, and $N_{\text{total}}$ is the total number of attacks conducted. The formula calculates the percentage of successful attacks, providing a quantitative measure of the attack's effectiveness.

## 4.2 COMPARATIVE EXPERIMENTAL RESULTS

In our experiments, we primarily focused on Image-Text Retrieval (ITR) tasks. We generated adversarial examples on various white-box models and then evaluated their effectiveness by calculating the attack success rates on both the white-box models and three additional black-box models.

Our analysis spanned two widely recognized datasets: Flickr30K, with a sample of 1,000 images and 5,000 captions, and MSCOCO, which provided a larger pool of 5,000 images and 25,000 captions. This broad dataset coverage allowed us to conduct a robust evaluation of our attack methods in image-text matching tasks, quantifying the success of adversarial examples in misleading these complex models. The detailed outcomes are methodically presented in TABLE 1 and TABLE 2.

Our results demonstrated that the OT-Attack method made significant strides in the creation of adversarial examples that were not only effective within models of the same type but also exhibited impressive cross-type attack success. This is particularly evident from the R@1 success rates in TR and IR tasks, where our adversarial examples maintained high effectiveness across varied models, including ALBEF, TCL, CLIP$_{ViT}$, and CLIP$_{CNN}$. For example, when using ALBEF to target TCL, our method improved the TR R@1 attack success rate by 6.95% on Flickr30K and 4.88% on MSCOCO, compared with the state-of-the-art results obtained by SGA. Conversely, in scenarios where TCL was employed to target ALBEF, our approach showed significant improvements over SGA, with increases of 8.41% on Flickr30K and 5.71% on MSCOCO in the TR R@1 attack success rate. The results demonstrate the effectiveness of improving adversarial transferability. Complementing our numerical analysis, Figure 4 (in Appendix) offers a visual representation of the impact of our adversarial examples. It contrasts the original images and texts with their modified versions, illuminating how subtle perturbations can drastically alter a model's performance in image-text matching tasks. The visual differences, particularly the nuanced texture changes introduced in the adversarial images, are made evident through difference masks, underscoring the deceptive potency of the adversarial examples and their potential to misguide VLP systems.

### 4.3 HYPERPARAMETER EXPERIMENTS

To more comprehensively demonstrate the superiority of our method, we conducted comparative experiments with the SGA across multiple sets of hyperparameters. For the sake of conciseness, we showcased the results using the ALBEF model as the source and TCL as the target, specifically focusing on the TR R@1 metric, with experiments conducted on the Flickr30K dataset.

#### 4.3.1 CAPTION QUANTITY

In our experiments on caption quantity, we evaluated the black-box attack success rates of our method versus the SGA in the context of image-text matching tasks across settings with caption quantities ranging from one to five. The dataset is Flickr30K. The source model is ALBEF and the target model is TCL. As demonstrated in TABLE 3, with an increase in the number of captions, there was a general trend of improvement in the Attack Success Rate (ASR), suggesting that a richer caption description leads to better attack efficacy. It is also evident that the OT-

Table 3: **ASR of experiments on caption quantity.**

| Attack | TR R@1 Caption Quantity | | | | |
|---|---|---|---|---|---|
| | 1 | 2 | 3 | 4 | 5 |
| SGA | 40.04 | 45.52 | 45.84 | 46.05 | 45.94 |
| OT-Attack (Ours) | **46.89** | **50.90** | **51.63** | **52.27** | **52.37** |

Attack outperformed the SGA across every caption quantity setting, indicating our method's superior performance across various caption quantities.

#### 4.3.2 SCALE QUANTITY

In our experiments concerning scale quantity, we examined the results of image set scaling at quantities of 1, 4, 5, and 7 (where 1 denotes no data augmentation, with only the original images being used for generating adversarial examples). The dataset is Flickr30K. The source model is ALBEF and the target model is TCL. As shown in TABLE 4, it is noteworthy that while the SGA's ASR decreased when the scale quantity increased to 7, the ASR of the OT-Attack continued to rise. Increasing the number of scales indeed improved the attack success rate, and our method's ASR was higher than that of the SGA across all

Table 4: **ASR of experiments on scale quantity.**

| Attack | TR R@1 Scale Quantity | | | |
|---|---|---|---|---|
| | 1 | 3 | 5 | 7 |
| SGA | 34.04 | 44.57 | 45.94 | 44.15 |
| OT-Attack (Ours) | **47.52** | **51.53** | **52.27** | **53.15** |

quantities. When the SGA's performance declined, the OT-Attack still showed an increase, demonstrating better robustness to variations in scale quantity.

Table 6: Adversarial Impact on Image Captioning Metrics.

| Attack | B@4 | METEOR | ROUGE-L | CIDEr | SPICE |
|---|---|---|---|---|---|
| Baseline | 39.7 | 31.0 | 60.0 | 133.3 | 23.8 |
| Co-Attack | 37.4 | 29.8 | 58.4 | 125.5 | 22.8 |
| SGA | 34.8 | 28.4 | 56.3 | 116.0 | 21.4 |
| OT-Attack (Ours) | **34.1** | **27.9** | **55.7** | **112.6** | **20.9** |

### 4.3.3 PERTURBATION STRENGTH

We also conducted experiments under different perturbation strengths, aiming to maintain imperceptibility to humans. We compared the results under three limited perturbation strengths: $\frac{2}{255}$, $\frac{4}{255}$, and $\frac{6}{255}$. The results are presented in TABLE 5. The dataset is Flickr30K. The source model is ALBEF and the target model is TCL. As the perturbation strength increased, both the SGA and OT-Attack experienced significant improvements in their ASR, with the OT-Attack consistently outperforming the SGA. Specifically, at a perturbation strength of $\frac{6}{255}$, the OT-Attack achieved an ASR of 90.20%. This demonstrates that the OT-Attack also exhibits superior performance as the perturbation strength increases. In the above hyperparameter experiments, the ASR of OT-Attack consistently surpassed that of SGA under identical experimental conditions. This comprehensively demonstrates the stability of OT-Attack across various hyperparameters.

Table 5: **ASR of experiments on perturbation strength.**

| Attack | TR R@1 Perturbation Strength | | |
|---|---|---|---|
| | $\frac{2}{255}$ | $\frac{4}{255}$ | $\frac{6}{255}$ |
| SGA | 45.42 | 72.81 | 82.72 |
| OT-Attack (Ours) | **52.37** | **82.93** | **90.20** |

## 4.4 CROSS-TASK TRANSFERABILITY

### 4.4.1 IMAGE CAPTIONING

In our research, we generated adversarial examples using the ALBEF model (Li et al., 2022) targeting the BLIP framework in a white-box scenario. BLIP is recognized for its advanced multi-modal encoder-decoder structure, which is trained on a diverse dataset with synthetic captions and noise reduction techniques. Our experiments were conducted on the MSCOCO dataset, examining both original and adversarially altered images. To evaluate the impact of our adversarial actions, we utilized a set of metrics designed for image captioning tasks, including BLEU (Papineni et al., 2002), METEOR (Banerjee et al., 2005), ROUGE (Lin, 2004), CIDEr (Vedantam et al., 2015), and SPICE (Anderson et al., 2016). These metrics assess various aspects of caption quality, from precision and semantic accuracy to recall, uniqueness, relevance, and the depiction of semantic properties.

The metrics used in our study offer varied insights into the text quality and relevance, giving a rounded view of the adversarial effects, as shown in TABLE 6. Our approach, compared to SGA, demonstrated lower scores across metrics: BLEU-4 decreased by 0.7, METEOR by 0.5, ROUGE-L by 0.6, CIDEr by 3.4, and SPICE by 0.5. These score reductions suggest our method's higher cross-task attack efficacy, with more significant decreases indicating better performance.

Figure 3 visually compares experimental outcomes, showing original versus adversarial image-caption pairs. These comparisons starkly exhibit how minor perturbations can drastically alter the model's interpretation, deviating from the intended meaning, thus highlighting our findings' practical significance.

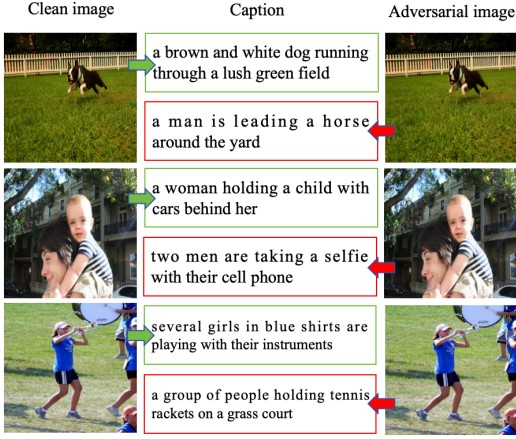

Figure 3: Comparison of Clean and Adversarial Image Captions.

Table 7: Performance on Visual Grounding Task Across RefCOCO+ Subsets.

| Attack | Val | TestA | TestB |
|--------|-----|-------|-------|
| Baseline | 58.4 | 65.9 | 46.2 |
| SGA | 56.5 | 63.7 | 45.4 |
| OT-Attack (Ours) | **56.3** | **63.5** | **45.0** |

Further delving into the realm of large-scale models, our experiments were conducted with specific parameters to gauge the extent of adversarial impact. We set the perturbation intensity at a subtle yet effective level of 16/255 and ran our adversarial process for 500 iterations. To assess the broader applicability and effectiveness of our attacks, we tested them on advanced models like GPT-4 and Bing Chat, posing the query "Describe this image" to these systems. The findings, illustrated in Figure 5 (in Appendix), reveal a notable level of success in our adversarial attacks, with these sophisticated models showing susceptibility to being misled.

### 4.4.2 VISUAL GROUNDING

To thoroughly evaluate the effectiveness of our adversarial attack strategies, we employed the RefCOCO+ (Yu et al., 2016) dataset, which is specifically curated for visual grounding tasks. This dataset comprises various subsets designed to evaluate different aspects of model performance, including:

- **RefCOCO+ val**: Offers a broad range of scenarios for a comprehensive evaluation.

- **RefCOCO+ testA**: Focuses on the model's ability to identify and localize human figures, testing its precision in distinguishing and positioning human subjects within images.

- **RefCOCO+ testB**: Targets the model's efficacy in recognizing and localizing non-human elements such as inanimate objects, animals, and various environmental features, challenging the model's versatility beyond human-centric tasks.

By leveraging the diverse testing scenarios provided by RefCOCO+, we aim to demonstrate the broad adaptability and transferability of our method across a wide array of visual grounding challenges, highlighting its potential for robust performance in varied contexts.

The quantitative analysis in TABLE 7 evaluates our adversarial examples' effectiveness against the ALBEF model, using TCL as the source. The baseline scores, representing unmodified samples, set the study's benchmark. Our OT-Attack strategy outperformed SGA, decreasing ALBEF's scores by 0.2 in Val, 0.2 in TestA, and 0.3 in TestB, evidencing our method's superior disruption of visual grounding. Additionally, in Figure 6 (in Appendix), visual analysis using the Flickr30K dataset demonstrates how minor perturbations significantly impair object recognition and localization in the ALBEF model, highlighting the impact of adversarial attacks on model accuracy and reliability.

## 5 CONCLUSION

We propose an Optimal Transport-based Adversarial Attack, *dubbed* OT-Attack. The proposed OT-Attack formulates the features of image and text sets as two distinct distributions, leveraging optimal transport theory to identify the most efficient mapping between them. It utilizes their mutual similarity as the cost matrix. The derived optimal mapping guides the generation of adversarial examples, effectively improving adversarial transferability. Extensive experiments across diverse network architectures and datasets in image-text matching tasks demonstrate the superior performance of the proposed OT-Attack in terms of adversarial transferability. Significantly, our results also show that OT-Attack is also effective in cross-task attacks, including image captioning and visual grounding, and poses a considerable challenge to commercial models such as GPT-4 and Bing Chat, highlighting the evolving landscape of adversarial threats in advanced AI applications. This underscores the need for robust defenses against sophisticated attacks.

## ETHICS STATEMENT

This paper proposes an optimal transport-based adversarial Attack for the VLP models, which may potentially generate harmful texts and pose risks. However, like previous adversarial attack methods, the proposed method explores adversarial perturbations with the goal of uncovering vulnerabilities in the VLP models. This effort aims to guide future work in enhancing the adversarial defense of the VLP models. Besides, the victim VLP models used in this paper are open-source models with publicly available weights. The research on adversarial attack and defense will collaboratively shape the landscape of AI security.

## REPRODUCIBILITY STATEMENT

We provide the source code for our OT-Attack in the supplementary materials. We will make the code publicly available after the work is accepted. The pseudocode for the proposed OT-Attack is shown in Appendix A and B. Experiment settings are reported in Section 4.1 in the submitted manuscript.

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

## A    SINKHORN ALGORITHM FOR OT

The Sinkhorn algorithm iteratively normalizes the rows and columns of the transport matrix to satisfy the marginal constraints while minimizing the regularized objective function (Cuturi, 2013). Here, $H(\mathbf{T})$ is the entropy of the transport matrix, introducing regularization (controlled by $\lambda$) to ensure numerical stability and efficient computation. Regarding the computation of Sinkhorn, the algorithm of the proposed OT-Attack is summarized in Algorithm 1.

## B    ALGORITHM OF ADVERSARIAL IMAGE GENERATION PROCESS

We employed the adversarial example generation method outlined in Equation 8 to create adversarial examples. These samples were then used to mount attacks on black-box models. The process is in Algorithm 2.

**Algorithm 1** Sinkhorn Iteration for OT

**Require:** $K$: cost matrix, $u$: source measure, $v$: target measure
**Ensure:** $T$: transport matrix
1: $r \leftarrow$ ones_like($u$)
2: $c \leftarrow$ ones_like($v$)
3: $thresh \leftarrow 1e - 2$
4: **for** $i = 1, \ldots, 100$ **do**
5:      $r_0 \leftarrow r$
6:      $r \leftarrow u/(\text{MatMul}(K, c))$
7:      $c \leftarrow v/(\text{MatMul}(K^\top, r))$
8:      $err \leftarrow \text{Mean}(\text{Abs}(r - r_0))$
9:      **if** $err < thresh$ **then**
10:         **break**
11:      **end if**
12: **end for**
13: $T \leftarrow \text{Outer}(r, c) \times K$
14: **return** $T$

---

**Algorithm 2** Adversarial Image Generation Process

**Require:** $model$: source model, $imgs$: original images, $\alpha$: adjustment factors, $X_{txt}$: textual representations
**Ensure:** $I_{adv}$: generated adversarial images
1: $model \rightarrow eval()$
2: $I_{adv} \leftarrow \text{clamp}(imgs.detach() + \text{Uniform}(-\epsilon, \epsilon), 0.0, 1.0)$
3: **for** each $i \in \{1 \ldots N\}$ **do**
4:      **for** $img \in I_{adv}$ **do**
5:         Apply data augmentations to $img$
6:         Extract features using $model$ on the augmented $img$
7:         Choose corresponding $X_{txt}$
8:         Calculate similarity and Wasserstein distance
9:         Optimize using Sinkhorn algorithm to find $T$
10:         Backpropagate using $loss_{OT}$ and update $img$
11:         $I'_{adv} \leftarrow \text{clamp}(I_{adv} + \text{sign}(\nabla_{img} loss), -\epsilon, \epsilon)$
12:         $I_{adv} \leftarrow I'_{adv}$
13:      **end for**
14: **end for**
15: **return** $I_{adv}$

## C   VISUALIZATION

### C.1   VISUALIZATION OF ADVERSARIAL EXAMPLES FROM FLICKR30K

Complementing our numerical analysis, Figure 4 offers a visual representation of the impact of our adversarial examples. It contrasts the original images and texts with their modified versions, illuminating how subtle perturbations can drastically alter a model's performance in image-text matching tasks. The visual differences, particularly the nuanced texture changes introduced in the adversarial images, are made evident through difference masks, underscoring the deceptive potency of the adversarial examples and their potential to misguide VLP systems.

### C.2   IMPACT OF ADVERSARIAL ATTACKS ON GPT-4 AND BING CHAT DESCRIPTIONS

Figure 5 reveals a notable level of success in our adversarial attacks, with these sophisticated models showing susceptibility to being misled.

### C.3   VISUALIZATION RESULTS FOR VISUAL GROUNDING

Additionally, visual analysis using the Flickr30K dataset and depicted in Figure 6 demonstrates how minor perturbations significantly impair object recognition and localization in the ALBEF model, highlighting the impact of adversarial attacks on model accuracy and reliability.

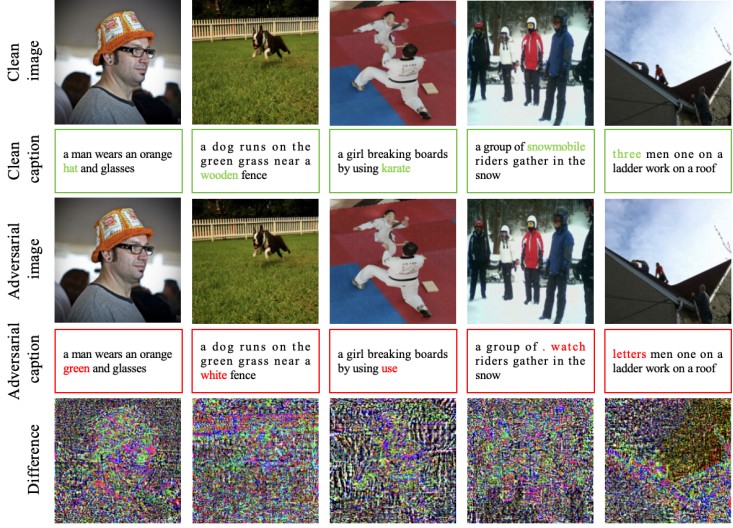

Figure 4: Visualization of adversarial examples from Flickr30K. In the task of image-text matching, adversarial examples for both images and texts were generated and utilized for image-to-text and text-to-image matching tasks, respectively. We have highlighted the distinctions in the text adversarial examples compared to the original samples and also quantified the pixel differences between the image adversarial examples and the original images.

## D   COMPARISON WITH MORE BASELINES

We compare the proposed OT-Attack with VLAttack (Yin et al., 2024), which focuses on enhancing the transferability of attacking pretrained vision-language models. ALBEF is employed as the source model in our experiments, and TCL is used as the target model. The results are shown in Table 8. The results show that our OT-Attack outperforms VLAttack (Yin et al., 2024) across all metrics, with notable improvements such as 52.37% vs. 43.2% in TR@1 and 61.05% vs. 52.04% in IR@1, demonstrating the superiority of our method in improving the adversarial transferability.

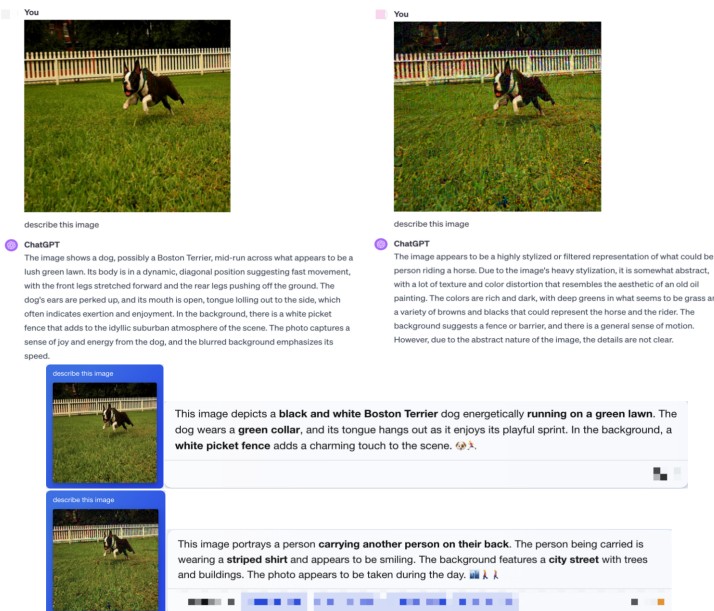

Figure 5: Impact of Adversarial Attacks on GPT-4 and Bing Chat Descriptions. This figure showcases the alterations in image descriptions by GPT-4 and Bing Chat before and after adversarial attacks. Original descriptions are compared to those generated from manipulated images, with increased perturbation strength and iteration count to mislead the AI models. The stark contrast in the outputs highlights the susceptibility of these models to adversarial examples, reflecting the effectiveness of the perturbations in altering the perceived content of the images.

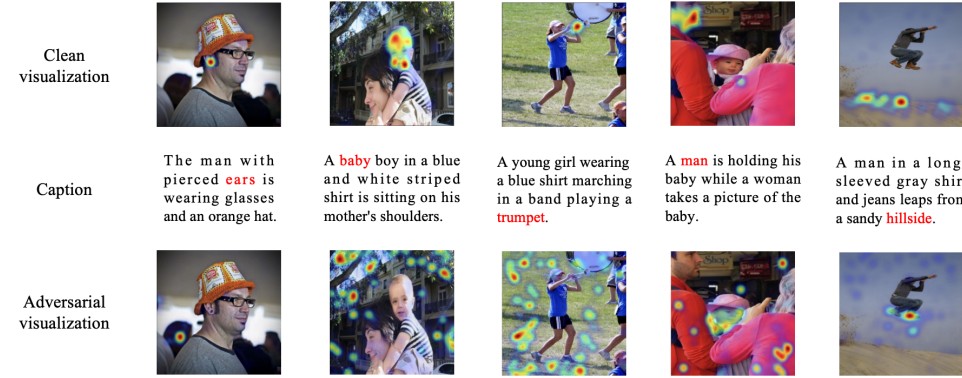

Figure 6: Visualization results for Visual Grounding. We employed TCL as the source model and ALBEF as the target model, with images and captions sourced from the Flickr30K dataset. The adversarial examples exhibit limited visual differences from the original samples; however, they disrupt the model's judgment of visual elements in the Visual Grounding task. Compared to clean data, the localization results for the same elements may have shifted or dispersed. The visualizations of Visual Grounding vividly demonstrate the disruptive impact of adversarial examples on the model.

Table 8: Comparative experimental results with VLAttack (Yin et al., 2024) on the Flickr30K dataset. The number in bold indicates the best jailbreak performance.

| Attack | TR @1 | TR @5 | TR @10 | IR @1 | IR @5 | IR @10 |
|---|---|---|---|---|---|---|
| VLAttack (Yin et al., 2024) | 43.2 | 23.09 | 16.01 | 52.04 | 32.14 | 24.21 |
| OT-Attack (ours) | **52.37** | **30.45** | **23.05** | **61.05** | **41.95** | **32.68** |

# E  COMPUTATIONAL COST

Following the default setting of SGA (Lu et al., 2023), we also adopt 1000 images from the Flickr30K dataset for our experiments. To evaluate performance, we compute the computational cost (minutes) and compare the OT-Attack with SGA across four models (ALBEF, TCL, $\text{CLIP}_{\text{ViT}}$, and $\text{CLIP}_{\text{CNN}}$). The results are shown in Table 9. OT-Attack consistently requires more time than SGA, which is approximately 1.75 times that of SGA, reflecting the added complexity of the proposed method.

Table 9: Computational cost (minutes) compared with SGA.

| Method | **ALBEF** | **TCL** | **$\text{CLIP}_{\text{ViT}}$** | **$\text{CLIP}_{\text{CNN}}$** |
|---|---|---|---|---|
| SGA | 34 | 33 | 17 | 13 |
| OT-Attack (ours) | 60 | 58 | 30 | 23 |

# F  HYPERPARAMETER ANALYSIS IN OT-ATTACK

The experimental settings of OT-Attack adhered to the default configurations in PLOT. This choice was primarily due to the insensitivity of OT-Attack to experimental parameters, as we will validate through ablation studies. In these studies, we independently evaluate the impact of three parameters $\lambda$, convergence threshold (thresh), and the iteration limit of the Sinkhorn algorithm—on the experimental outcomes. ALBEF is used as the source model, and TCL as the target model.

## F.1  SENSITIVITY ANALYSIS OF $\lambda$

The parameter $\lambda$ balances the minimization of transport cost and plan smoothness. In our experiments, the default value was set to 0.1. To analyze the sensitivity of $\lambda$, we conduct OT-Attack with different $\lambda$. The results are shown in Table 10. The results indicate that varying $\lambda$ among 0.01, 0.1, and 1 while keeping other conditions constant leads to consistent attack success rates across metrics such as TR R@1, TR R@5, TR R@10, IR R@1, IR R@5, and IR R@10. This demonstrates that OT-Attack is robust to changes in $\lambda$, maintaining stable performance.

Table 10: Performance of the proposed OT-Attack with different $\lambda$ values.

| $\lambda$ Value | TR @1 | TR @5 | TR @10 | IR @1 | IR @5 | IR @10 |
|---|---|---|---|---|---|---|
| 0.01 | 52.27 | 30.35 | 22.95 | 60.98 | 41.80 | 32.86 |
| 0.1 | 52.37 | 30.45 | 23.05 | 61.00 | 41.95 | 32.68 |
| 1.0 | 52.90 | 30.45 | 23.05 | 61.17 | 41.95 | 32.68 |

## F.2  CONVERGENCE THRESHOLD AND SINKHORN ITERATION LIMIT

The convergence threshold (thresh) and the iteration limit of the Sinkhorn algorithm are strongly interdependent. The convergence of the Sinkhorn algorithm ensures that the transport matrix $P = \text{diag}(r) \cdot K \cdot \text{diag}(c)$ satisfies the prescribed marginal distributions $u$ and $v$. Convergence is typically assessed by measuring the change in the scaling factors $r$ or $c$ between successive iterations, where the error metric (e.g., $\|r^{(k)} - r^{(k-1)}\|$) must fall below a predefined threshold $\epsilon$. Alternatively, convergence can be determined by the deviation of the row and column sums of $P$ from $u$ and $v$. A maximum iteration limit is often imposed to prevent infinite loops.

In this study, the default settings were thresh $= 1.00 \times 10^{-2}$ and an iteration limit of 100. First, we analyzed the average number of iterations under thresh $= 1.00 \times 10^{-2}$, finding that the mean iteration count for generating 1,000 adversarial samples was only 2.3. Further, we examined the error scalar after each iteration. The error scalar starts at the order of $1.00 \times 10^3$ after the first iteration, reaches $1.00 \times 10^{-2}$ within two iterations, and decreases to $1.00 \times 10^{-3}$ or $1.00 \times 10^{-4}$ after three iterations. This analysis indicates that thresh should range between $1.00 \times 10^3$ and $1.00 \times 10^{-6}$. If thresh exceeds $1.00 \times 10^3$, the Sinkhorn algorithm converges in just one iteration.

We also conduct OT-Attack with different threshold values. The results are shown in Table 11. It reveals minimal differences in image-text matching metrics when the source model is ALBEF, and the target model is TCL, confirming that OT-Attack is insensitive to thresh and demonstrates good stability. Additionally, We also conduct OT-Attack with different iteration limit values. The results are shown in Table 12. It demonstrates that the attack efficacy of OT-Attack remains stable.

Table 11: Performance of the proposed OT-Attack with different threshold values.

| Threshold Value | TR @1 | TR @5 | TR @10 | IR @1 | IR @5 | IR @10 |
|---|---|---|---|---|---|---|
| $1.00 \times 10^{-6}$ | 53.32 | 30.45 | 22.85 | 61.12 | 41.82 | 33.04 |
| $1.00 \times 10^{-2}$ | 52.37 | 30.45 | 23.05 | 61.00 | 41.95 | 32.68 |
| $1.00 \times 10^{3}$ | 53.11 | 30.85 | 23.05 | 60.86 | 41.64 | 32.80 |

Table 12: Performance of the proposed OT-Attack with different iteration limit values.

| Iteration Limit Value | TR @1 | TR @5 | TR @10 | IR @1 | IR @5 | IR @10 |
|---|---|---|---|---|---|---|
| 1 | 52.21 | 30.25 | 22.85 | 60.64 | 41.80 | 32.34 |
| 2 | 52.32 | 30.45 | 23.05 | 60.86 | 41.95 | 32.68 |
| 3 | 52.37 | 30.45 | 23.05 | 61.00 | 41.95 | 32.68 |
| 100 | 52.37 | 30.45 | 23.05 | 61.00 | 41.95 | 32.68 |

In summary, OT-Attack exhibits robustness to hyperparameter variations, delivering stable attack performance under different parameter settings, and significantly outperforms SGA. In this study, Sinkhorn convergence is determined based on the condition that the variation in the row normalization factor r between consecutive iterations is below the predefined threshold $\epsilon$, $i.e.$, $\|r^{(k)} - r^{(k-1)}\| < \epsilon$.

# G  EXPERIMENTS ON CHATGPT4 AND BING

We randomly sample 100 images from Flickr30K to generate adversarial examples by using SGA and our OT-Attack. Then we evaluate them on ChatGPT4 and Bing. The results are shown in Table 13. It highlight the superiority of our OT-Attack, achieving significantly higher ASR rates of 24% on ChatGPT4 and 30% on Bing, compared to only 7% and 8% by SGA.

Table 13: Performance of the proposed OT-Attack on ChatGPT4 and Bing.

| Models | ChatGPT4 | Bing |
|---|---|---|
| SGA | 7% | 8% |
| OT-Attack | **24%** | **30%** |

# H  ANALYSIS OF THE EFFECTIVENESS OF OT-ATTACK

Overfitting adversarial examples (AEs) to the source model can significantly reduce the attack transferability. To quantify the risk of overfitting, we leverage the PAC-Bayes theorem to measure the information stored in the network's weights (IIW) (Wang et al., 2022), a promising indicator of generalization ability. Lower IIW values indicate reduced overfitting risks. For each AE generated by SGA or our method during optimization iterations, we compute its IIWs by feeding it into four VLMs. We evaluate the IIWs of 1,000 AEs throughout the optimization process and present the averaged results in Figure 7. During optimization, the IIW of AEs from the SOTA baseline (SGA) initially decreases but then sharply rises. In contrast, our method maintains consistently low IIW values for generated AEs, effectively mitigating overfitting risks. Consequently, our method enhances attack transferability.

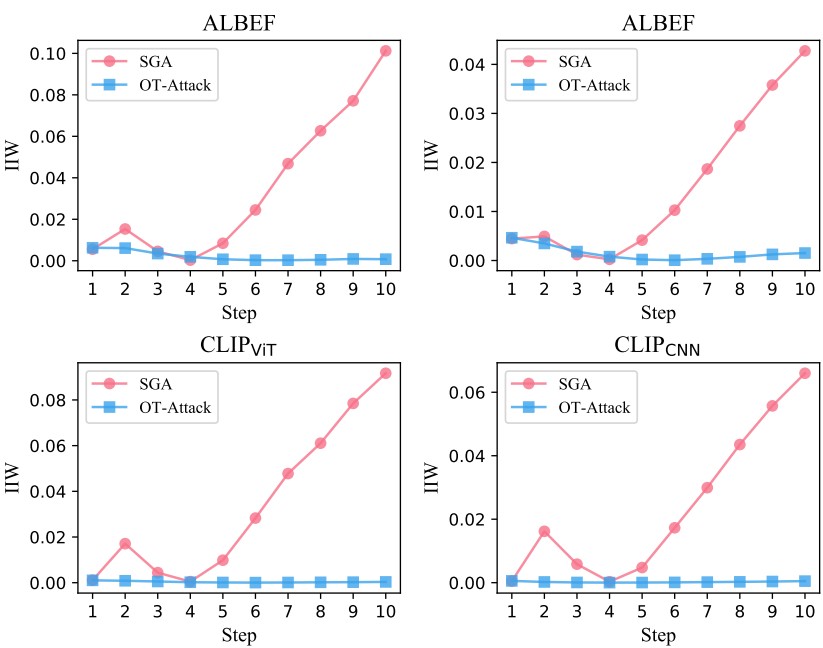

Figure 7: Overfitting analysis of SGA & our OT-Attack via IIWs.

## I  EXPERIMENTS ON MORE SOURCE MODELS

We adopt more vision-language models as the source models for experiments, such as Eva-CLIP and BLIP2, for image-text retrieval tasks. The results on Eva-CLIP are shown in Table 14. It highlights the superior performance of our OT-Attack compared to the baseline SGA method in adversarial attack success rates on image-text retrieval tasks. Across all models, OT-Attack achieves higher results in both TR R@1 and IR R@1 metrics. Notably, on challenging models like ALBEF and TCL, OT-Attack outperforms SGA by significant margins (e.g., IR R@1: 28.23 vs. 26.23 on ALBEF, 32.42 vs. 30.12 on TCL). Similarly, OT-Attack consistently achieves the best results for CLIP-based models, demonstrating its effectiveness across diverse architectures. The results on BLIP2 are shown in Table 15. It also highlights the effectiveness of OT-Attack, which consistently outperforms SGA across all models and metrics. Notably, OT-Attack achieves higher success rates on ALBEF (TR R@1: 58.89 vs. 51.23, IR R@1: 69.23 vs. 63.53) and TCL (TR R@1: 52.18 vs. 48.42, IR R@1: 64.27 vs. 58.94). Hence, our OT-Attack demonstrates superior adversarial sample transferability compared to SGA.

Table 14: Adversarial Attack Success Rates on Image-Text Retrieval. Eva-CLIP is used as the source model. The number in bold indicates the best attack performance.

| Models | Eva-CLIP | | ALBEF | | TCL | | CLIP$_{ViT}$ | | CLIP$_{CNN}$ | |
|---|---|---|---|---|---|---|---|---|---|---|
| | TR R@1 | IR R@1 | TR R@1 | IR R@1 | TR R@1 | IR R@1 | TR R@1 | IR R@1 | TR R@1 | IR R@1 |
| SGA | 99.23 | 99.1 | 12.98 | 26.23 | 16.01 | 30.12 | 40.12 | 51.44 | 35.79 | 46.79 |
| OT-Attack (ours) | 98.88 | 98.43 | **13.35** | **28.23** | **16.14** | **32.42** | **44.84** | **56.96** | **40.05** | **50.12** |

Table 15: Adversarial Attack Success Rates on Image-Text Retrieval. BLIP2 is used as the source model. The number in bold indicates the best attack performance.

| Models | BLIP2 | | ALBEF | | TCL | | CLIP$_{ViT}$ | | CLIP$_{CNN}$ | |
|---|---|---|---|---|---|---|---|---|---|---|
| | TR R@1 | IR R@1 | TR R@1 | IR R@1 | TR R@1 | IR R@1 | TR R@1 | IR R@1 | TR R@1 | IR R@1 |
| SGA | **98.56** | **98.67** | 51.23 | 63.53 | 48.42 | 58.94 | 35.17 | 45.01 | 36.32 | 48.08 |
| OT-Attack (ours) | 97.72 | 97.66 | **58.89** | **69.23** | **52.18** | **64.27** | **36.46** | **48.75** | **44.13** | **52.12** |

Table 16: Adversarial Impact on Image Captioning Metrics. The table demonstrates the effects of adversarial attacks on image captioning, using 10,000 images from MSCOCO and attacks generated via ALBEF, with captions by MiniGPT4 and Qwen2-VL. The evaluation employed metrics like BLEU-4, METEOR, ROUGE-L, CIDEr, and SPICE, where lower scores signify more impactful attacks. The number in bold indicates the best attack performance.

| MiniGPT4 | B@4 | METEOR | ROUGE-L | CIDEr | SPICE |
|---|---|---|---|---|---|
| Clean | 32.5 | 33.2 | 60.3 | 128.7 | 21.8 |
| SGA | 30.4 | 27.3 | 56.2 | 113.6 | 20.5 |
| OT-Attack (ours) | **30.1** | **26.7** | **54.8** | **109.5** | **20.3** |
| **Qwen2-VL** | **B@4** | **METEOR** | **ROUGE-L** | **CIDEr** | **SPICE** |
| Clean | 38.7 | 34.9 | 66.8 | 121.5 | 25.4 |
| SGA | 35.2 | 32.1 | 62.9 | 108.4 | 22.6 |
| OT-Attack (ours) | **34.9** | **31.4** | **61.6** | **103.9** | **21.7** |

## J  EXPERIMENTS ON MORE MODELS FOR IMAGE GENERATION TASKS

We adopt more models, such as MiniGPT4 and Qwen2-VL, for image generation tasks. We compare the impact of adversarial attacks on image captioning performance for MiniGPT4 and Qwen2-VL with the prompt "Describe the image" using metrics like BLEU-4, METEOR, ROUGE-L, CIDEr, and SPICE. The results are shown in Table 16. It is clear that our OT-Attack consistently outperforms SGA, achieving the lowest scores across most metrics, which indicates more effective attacks. For instance, OT-Attack reduces CIDEr scores to 109.5 for MiniGPT4 and 103.9 for Qwen2-VL, highlighting its superior ability to degrade captioning performance compared to other methods. These results underscore the efficacy of the proposed OT-Attack.

## K  ABLATION STUDY OF OUR OT-ATTACK

Table 17 presents the ablation study of our proposed OT-Attack, evaluating its adversarial attack success rates on image-text retrieval tasks. ALBEF is used as the source model. The study compares three settings: removing the optimal transport mechanism (OT-Attack w/o OT), removing data augmentation strategies (OT-Attack w/o Augmentation), and the complete method (OT-Attack). The results show that removing OT or augmentation can reduce the adversarial transferability. Notably, the complete OT-Attack achieves the best adversarial transferability, highlighting the critical role of optimal transport and data augmentation in the proposed OT-Attack.

Table 17: Ablation study of the proposed OT-Attack. Adversarial Attack Success Rates on Image-Text Retrieval. The number in bold indicates the best attack performance.

| Models | ALBEF | | TCL | | CLIP$_{ViT}$ | | CLIP$_{CNN}$ | |
|---|---|---|---|---|---|---|---|---|
| | TR R@1 | IR R@1 | TR R@1 | IR R@1 | TR R@1 | IR R@1 | TR R@1 | IR R@1 |
| OT-Attack w/o OT | **97.2** | **97.3** | 45.4 | 55.3 | 33.4 | 44.2 | 34.9 | 46.6 |
| OT-Attack w/o Augmentation | **97.2** | 96.9 | 46.3 | 55.2 | 33.4 | 44.1 | 39.3 | 51.2 |
| OT-Attack | 95.9 | 95.9 | **52.4** | **61.1** | **34.9** | **47.1** | **42.3** | **53.0** |

