# OpenReview forum: "OT-Attack: Enhancing Adversarial Transferability of Vision-Language Models via Optimal Transport Optimization"
_ICLR.cc/2025/Conference — ICLR 2025 Conference Withdrawn Submission_

### Official Review · Reviewer_yYgK · 2024-10-30

**Soundness:** 3
**Presentation:** 3
**Contribution:** 3
**Rating:** 6
**Confidence:** 3

**Summary:**

This paper investigates data augmentations in the context of transferable attacks on Vision-Language Processing (VLP) models. Compared to existing methods, the proposed approach achieves improved alignment between augmented images and text. Specifically, the authors leverage optimal transport theory to derive the most efficient mapping between augmented image distributions and text distributions. They conduct experiments on image-text retrieval, image captioning, and visual grounding tasks to demonstrate their improved transferability.

**Strengths:**

1. This paper identifies a key issue in existing attacks: a mismatch in augmentations applied to images versus text. To address this, the authors utilize optimal transport theory, enhancing transferability. The motivation and proposed solutions are logical and well-founded.
2. The authors conduct extensive experiments across four VLP models and multiple tasks, effectively demonstrating the superior performance of their approach.

**Weaknesses:**

My primary concern is the focus on attacking VLP models solely through adversarial image generation, while neglecting the equally crucial role of text in these models. Since texts play an integral part in VLP models, a comprehensive approach should consider adversarial perturbations for both modalities. For example, incorporating a text-to-image generation component could be beneficial to observe how adversarial texts lead to variations in generated images.

**Questions:**

1. What is the time cost associated with using optimal transport theory?
2. How are augmented texts generated?

---

> ### Author Response · Authors · 2024-11-22
> **Rebuttal by Authors**
>
> Thank you for your valuable review and suggestions. Below we respond to the comments in **Weaknesses (W)** and **Questions (Q)**.
>
> ***W1: My primary concern is the focus on attacking VLP models solely through adversarial image generation, while neglecting the equally crucial role of text in these models.***
>
> We sincerely appreciate the reviewer’s comprehensive consideration of multimodal attacks. Indeed, both SGA and OT-Attack target both text and image modalities. The primary approach involves using images to guide the generation of adversarial text and using text to guide the generation of adversarial images. OT-Attack focuses on applying optimal transport theory to the image modality, and the optimization of adversarial images inherently leads to improved results for adversarial text generation.
>
>
>
>
>
> ***Q1: What is the time cost associated with using optimal transport theory?***
>
> Thank you for the feedback. Following your suggestion, we compute the computational cost (minutes) and compare the OT-Attack with SGA across four models (ALBEF, TCL, CLIP_VIT , and CLIP_CNN ). The results are shown in the following Table. Our OT-Attack consistently requires more time than SGA, which is approximately 1.75 times that of SGA, reflecting the added complexity of the proposed method.  Please refer to **Appendix E** for more details.
>
>
>
> | Method         | ALBEF | TCL | CLIP_VIT | CLIP_CNN |
> |----------------|-------|-----|----------|----------|
> | SGA           | 34    | 33  | 17       | 13       |
> | OT-Attack (ours) | 60    | 58  | 30       | 23       |
>
>
> ***Q2: How are augmented texts generated?***
>
> We sincerely appreciate the reviewer's attention to text augmentation. In the current study, we adhered to the five standard captions provided by the dataset to ensure experimental comparability and did not perform text augmentation. However, more detailed textual descriptions could be generated by leveraging VLP models such as MiniGPT-V, LLaVA, or InstructBLIP to describe images from multiple perspectives or simply through synonym substitution. We believe that more comprehensive textual information could enhance adversarial samples' transferability, providing a valuable direction for future research.

---

> ### Author Response · Authors · 2024-11-24
> **Official Comment by Authors**
>
> Dear Reviewer yYgK,
>
> Thank you once again for taking the time to review our paper. We deeply value your insightful comments and are sincerely grateful for your efforts.
>
> We kindly request you to review our reply to see if it sufficiently addresses your concerns. Your feedback means a lot to us.
>
> Thank you deeply, Authors

---

> > ### Comment · Reviewer_yYgK · 2024-11-24
> > **Response by Reviewer**
> >
> > Thank you for your response; you addressed some of my concerns.
> > However, the experiments in Appendix E indicate that the proposed method lacks efficiency. And based on other comments, I would prefer to maintain my score.

---

> > > ### Author Response · Authors · 2024-11-25
> > > **Thank you for your support**
> > >
> > > Thank you for your support ! We greatly appreciate your insightful feedback and suggestions.

---

### Official Review · Reviewer_vFsY · 2024-10-31

**Soundness:** 3
**Presentation:** 2
**Contribution:** 2
**Rating:** 5
**Confidence:** 4

**Summary:**

In this paper, the authors propose a transfer-based adversarial attack method for vision-language pre-training (VLP) models that leverages image-text augmentations and optimal transport theory to enhance adversarial example transferability. Experiments are conducted across various vision-language tasks and VLP models.

**Strengths:**

1. This paper aims to enhance the transferability of adversarial examples for vision-language pre-training models, addressing a practical and increasingly relevant issue as these models gain importance in real-world applications.
2.  Adversarial examples generated by OT-Attack are tested on commercial models like GPT-4 and BingChat.

**Weaknesses:**

1. Overall, the manuscript presents valuable insights; however, enhancing the clarity and coherence of the writing, particularly in the methods section, would significantly improve its readability and impact.
2. The titles of Tables 1 and 2 are identical, which may easily confuse readers and hinder their understanding of the experiments. Differentiating the titles would improve clarity and facilitate comprehension of the authors' work.

**Questions:**

1. Co-attack and SGA were proposed prior to 2023, and since then, many new vision-language models have emerged. To further validate the effectiveness of the proposed method, the authors should evaluate additional vision-language models, such as Eva-CLIP and BLIP2 for image-text retrieval tasks, and MiniGPT4 and Qwen2-VL for image generation tasks in a transfer-based setting.
2. I did not find any ablation experiments addressing the role of image-text augmentation and optimal transport theory in OT-attack.
3. The authors claim that optimal transport can enhance adversarial example transferability; I recommend providing a theoretical explanation and further experimental analysis to support this assertion.

---

> ### Author Response · Authors · 2024-11-22
> **Rebuttal by Authors [1/3]**
>
> Thank you for your valuable review and suggestions. Below we respond to the comments in **Weaknesses (W)** and **Questions (Q)**.
>
>
> ***W1: Enhancing the clarity and coherence of the writing, particularly in the methods section, would significantly improve its readability and impact.***
>
> Thank you for your feedback. We have revised the methods section to enhance its clarity and coherence, improving readability and impact.  Please refer to **Approach Section** of Page 3-5.
>
>
> ***W2: The titles of Tables 1 and 2 are identical, which may easily confuse readers and hinder their understanding of the experiments.***
>
> Thank you for pointing this out. We rewrite the captions of Table 1 and Table 2 so that readers can distinguish them and improve readability. Please refer to Table 1 of Page 6 and Table 2 of Page 7.

---

> ### Author Response · Authors · 2024-11-22
> **Rebuttal by Authors [2/3]**
>
> ***Q1: The authors should evaluate additional vision-language models, such as Eva-CLIP and BLIP2 for image-text retrieval tasks, and MiniGPT4 and Qwen2-VL for image generation tasks in a transfer-based setting.***
>
>
> Thank you for your feedback. We adopt more vision-language models as the source models for experiments, such as Eva-CLIP and BLIP2, for image-text retrieval tasks. The results on Eva-CLIP are shown in the following Table.
>
> | Models           | Eva-CLIP (TR R@1) | Eva-CLIP (IR R@1) | ALBEF (TR R@1) | ALBEF (IR R@1) | TCL (TR R@1) | TCL (IR R@1) | CLIP_VIT (TR R@1) | CLIP_VIT (IR R@1) | CLIP_CNN (TR R@1) | CLIP_CNN (IR R@1) |
> |------------------|--------------------|--------------------|----------------|----------------|--------------|--------------|-------------------------|-------------------------|-------------------------|-------------------------|
> | SGA              | 99.23             | 99.1              | 12.98          | 26.23          | 16.01        | 30.12        | 40.12                  | 51.44                  | 35.79                  | 46.79                  |
> | OT-Attack (ours) | 98.88             | 98.43             | **13.35**          | **28.23**          | **16.14**        | **32.42**       | **44.84**                  | **56.96**                  | **40.05**                  | **50.12**                  |
>
> It highlights the superior performance of our OT-Attack compared to the baseline SGA method in adversarial attack success rates on image-text retrieval tasks. Across all models, OT-Attack achieves higher results in both TR R@1 and IR R@1 metrics. Notably, on challenging models like ALBEF and TCL, OT-Attack outperforms SGA by significant margins (e.g., IR R@1: 28.23 vs. 26.23 on ALBEF, 32.42 vs. 30.12 on TCL). Similarly, OT-Attack consistently achieves the best results for CLIP-based models, demonstrating its effectiveness across diverse architectures.
>
> The results on BLIP2 are shown in the following Table. It also highlights the effectiveness of OT-Attack, which consistently outperforms SGA across all models and metrics. Notably, OT-Attack achieves higher success rates on ALBEF (TR R@1: 58.89 vs. 51.23, IR R@1: 69.23 vs. 63.53) and TCL (TR R@1: 52.18 vs. 48.42, IR R@1: 64.27 vs. 58.94). Hence, our OT-Attack demonstrates superior adversarial sample transferability compared to SGA. Please refer to **Appendix I** for more details.
>
> | Models           | BLIP2 (TR R@1) | BLIP2 (IR R@1) | ALBEF (TR R@1) | ALBEF (IR R@1) | TCL (TR R@1) | TCL (IR R@1) | CLIP_VIT (TR R@1) | CLIP_VIT (IR R@1) | CLIP_CNN (TR R@1) | CLIP_CNN (IR R@1) |
> |------------------|----------------|----------------|----------------|----------------|--------------|--------------|-------------------------|-------------------------|-------------------------|-------------------------|
> | SGA              | 98.56          | 98.67          | 51.23          | 63.53          | 48.42        | 58.94        | 35.17                  | 45.01                  | 36.32                  | 48.08                  |
> | OT-Attack (ours) | 97.72          | 97.66          | **58.89**          | **69.23**          | **52.18**        | **64.27**        | **36.46**                  | **48.75**                  | **44.13**                  | **52.12**                  |
>
> We adopt more models, such as MiniGPT4 and Qwen2-VL, for image generation tasks. We compare the impact of adversarial attacks on image captioning performance for MiniGPT4 and Qwen2-VL with the prompt ``Describe the image`` using metrics like BLEU-4, METEOR, ROUGE-L, CIDEr, and SPICE. The results are shown in the following Table. It is clear that our OT-Attack consistently outperforms SGA, achieving the lowest scores across most metrics, which indicates more effective attacks. For instance, OT-Attack reduces CIDEr scores to 109.5 for MiniGPT4 and 103.9 for Qwen2-VL, highlighting its superior ability to degrade captioning performance compared to other methods. These results underscore the efficacy of the proposed OT-Attack. Please refer to **Appendix J** for more details.
>
> | Model         | Method       | B@4  | METEOR | ROUGE-L | CIDEr  | SPICE |
> |---------------|--------------|-------|--------|---------|--------|-------|
> | MiniGPT4      | Clean        | 32.5  | 33.2   | 60.3    | 128.7  | 21.8  |
> | MiniGPT4      | SGA          | 30.4  | 27.3   | 56.2    | 113.6  | 20.5  |
> | MiniGPT4      | OT-Attack    | **30.1**  | **26.7**   | **54.8**    | **109.5**  | **20.3**  |
> | Qwen2-VL      | Clean        | 38.7  | 34.9   | 66.8    | 121.5  | 25.4  |
> | Qwen2-VL      | SGA          | 35.2  | 32.1   | 62.9    | 108.4  | 22.6  |
> | Qwen2-VL      | OT-Attack    | **34.9**  | **31.4**   | **61.6**    | **103.9**  | **21.7**  |

---

> ### Author Response · Authors · 2024-11-22
> **Rebuttal by Authors [3/3]**
>
> ***Q2: I did not find any ablation experiments addressing the role of image-text augmentation and optimal transport theory in OT-attack.***
>
> Thank you for pointing this out. We conduct an  ablation study of the proposed OT-Attack. ALBEF is used as the source model. The study compares three settings: removing the optimal transport mechanism (OT-Attack w/o OT), removing data augmentation strategies (OT-Attack w/o Augmentation), and the complete method (OT-Attack). The results are shown in the following table. The results show that removing OT or augmentation can reduce the adversarial transferability. Notably, the complete OT-Attack achieves the best adversarial transferability, highlighting the critical role of optimal transport and data augmentation in the proposed OT-Attack.  Please refer to **Appendix K** for more details.
>
> | Models                     | ALBEF (TR R@1) | ALBEF (IR R@1) | TCL (TR R@1) | TCL (IR R@1) | CLIP_VIT (TR R@1) | CLIP_VIT (IR R@1) | CLIP_CNN (TR R@1) | CLIP_CNN (IR R@1) |
> |----------------------------|----------------|----------------|--------------|--------------|-------------------------|-------------------------|-------------------------|-------------------------|
> | OT-Attack w/o OT           | **97.2**       | **97.3**       | 45.4         | 55.3         | 33.4                   | 44.2                   | 34.9                   | 46.6                   |
> | OT-Attack w/o Augmentation | **97.2**       | 96.9           | 46.3         | 55.2         | 33.4                   | 44.1                   | 39.3                   | 51.2                   |
> | OT-Attack                  | 95.9           | 95.9           | **52.4**     | **61.1**     | **34.9**               | **47.1**               | **42.3**               | **53.0**               |
>
> ***Q3: I recommend providing a theoretical explanation and further experimental analysis to support this assertion.***
>
>
> Thank you for your feedback. Overfitting adversarial examples (AEs) to the source model can significantly reduce the attack transferability. To quantify the risk of overfitting, we leverage the PAC-Bayes theorem to measure the information stored in the network's weights (IIW), a promising indicator of generalization ability. Lower IIW values indicate reduced overfitting risks. For each AE generated by SGA or our method during optimization iterations, we compute its IIWs by feeding it into four VLMs. We evaluate the IIWs of 1,000 AEs throughout the optimization process and present the averaged results in $\\textrm{\\color{blue}Figure 7}$ (Page 19). During optimization, the IIW of AEs from the SOTA baseline (SGA) initially decreases but then sharply rises. In contrast, our method maintains consistently low IIW values for generated AEs, effectively mitigating overfitting risks. Consequently, our method enhances attack transferability.  Please refer to **Appendix H** for more details.

---

> ### Author Response · Authors · 2024-11-24
> **Official Comment by Authors**
>
> Dear Reviewer vFsY,
>
> Thank you once again for taking the time to review our paper. We deeply value your insightful comments and are sincerely grateful for your efforts.
>
> We kindly request you to review our reply to see if it sufficiently addresses your concerns. Your feedback means a lot to us.
>
> Thank you deeply, Authors

---

> > ### Comment · Reviewer_vFsY · 2024-11-25
> >
> > Thank you for your response. I still have a few concerns:
> >
> > 1. This paper focuses on the transferability of multimodal adversarial examples. I suggest evaluating additional vision-language models, such as Eva-CLIP and BLIP2. However, the use of Eva-CLIP and BLIP2 as source models to evaluate smaller VLP (e.g. ALBEF, TCL) models is unclear and somewhat confusing.
> >
> > 2. Unlike BLIP2, MiniGPT4 and Qwen2-VL generate descriptions of varying lengths based on the `max_new_tokens` setting. MiniGPT4 may produce incomplete captions if `max_new_tokens` is set too low, while longer descriptions can significantly impact metrics like BLEU and METEOR. Providing more details on this issue would be beneficial.

---

> > > ### Author Response · Authors · 2024-11-25
> > > **Rebuttal by Authors**
> > >
> > > ***1: This paper focuses on the transferability of multimodal adversarial examples. I suggest evaluating additional vision-language models, such as Eva-CLIP and BLIP2. However, the use of Eva-CLIP and BLIP2 as source models to evaluate smaller VLP (e.g. ALBEF, TCL) models is unclear and somewhat confusing.***
> > >
> > > Thank you for your feedback. We also adopt the Eva-CLIP and BLIP2 as the target model to conduct experiments. Eva-CLIP shares a similar structure with CLIP_ViT and CLIP_CNN mentioned in the manuscript, while BLIP2 is structurally comparable to ALBEF and TCL. We adopt ALBEF and CLIP_ViT as the source models to conduct experiments. The results are shown in the following Table.  It is clear that the proposed OT-Attack achieves significant improvements over SGA on BLIP2 (TR R@1: 48.55 vs. 40.67, IR R@1: 51.24 vs. 47.32) and Eva-CLIP (TR R@1: 34.02 vs. 32.18, IR R@1: 46.43 vs. 42.96). These results highlight OT-Attack's superior performance.
> > >
> > > | Source Model | Attack     | Target Model | ALBEF TR R@1 | ALBEF IR R@1 | TCL TR R@1 | TCL IR R@1 | BLIP2 TR R@1 | BLIP2 IR R@1 | CLIP_ViT TR R@1 | CLIP_ViT IR R@1 | CLIP_CNN TR R@1 | CLIP_CNN IR R@1 | Eva-CLIP TR R@1 | Eva-CLIP IR R@1 |
> > > |--------------|------------|--------------|--------------|--------------|------------|------------|--------------|--------------|----------------|----------------|----------------|----------------|----------------|----------------|
> > > | ALBEF        | SGA        |              | 97.24        | 97.28        | 45.42      | 55.25      | 40.67        | 47.32        | 33.38          | 44.16          | 34.93          | 46.57          | 32.18          | 42.96          |
> > > | ALBEF        | OT-Attack  |              | 95.93        | 95.86        | 52.37      | 61.05      | 48.55        | 51.24        | 34.85          | 47.10           | 42.33          | 50.03          | 34.02          | 46.43          |
> > > | CLIP_ViT     | SGA        |              | 13.40         | 27.22        | 16.23      | 30.76      | 12.66        | 24.97        | 99.08          | 98.94          | 38.76          | 47.79          | 42.98          | 54.16          |
> > > | CLIP_ViT     | OT-Attack  |              | 14.29        | 29.28        | 16.58      | 33.49      | 13.23        | 26.01        | 98.65          | 98.52          | 43.55          | 50.50           | 50.73          | 60.32          |
> > >
> > > ***2:Unlike BLIP2, MiniGPT4 and Qwen2-VL generate descriptions of varying lengths based on the max_new_tokens setting. MiniGPT4 may produce incomplete captions if max_new_tokens is set too low, while longer descriptions can significantly impact metrics like BLEU and METEOR. Providing more details on this issue would be beneficial.***
> > >
> > > Thank you for your suggestion. Indeed, we should provide more detailed descriptions of the experimental settings. In the experiments with MiniGPT and Qwen2-VL, we set     ``max_new_tokens ``=1000 to allow for complete outputs of meaningful content. The models were prompted with  ``Describe the image ``. In the generated results, we abbreviated any infinite repetitions up to the stop token, removed all emojis, and retained only the English text.

---

> > > ### Author Response · Authors · 2024-12-02
> > > **Looking forward to further feedback**
> > >
> > > Dear Reviewer vFsY,
> > >
> > > We appreciate the time and effort you have dedicated to providing insightful review. If there are any additional clarifications or information needed from our side, please let us know. Thank you again for your valuable insights, and we look forward to any updates you may have.
> > >
> > > Best,
> > > The Authors

---

### Official Review · Reviewer_uWjN · 2024-11-03

**Soundness:** 3
**Presentation:** 2
**Contribution:** 3
**Rating:** 8
**Confidence:** 3

**Summary:**

This paper proposes a new adversarial attack for vision-language models called OT-Attack, pointing out that the existing SGA attack has a limitation in considering correct image-text matching after augmentations.

OT-Attack utilizes Optimal Transport to capture valid image-text matching between augmented images and captions (calculate transportation matrix $T_{ij}$). Then, adversarial attack optimization maximizes the $loss_{OT}$ ($\sum T_{ij} C_{ij}$) to update an adversarial image.

The method improves the transferability of the SGA attack in three tasks: image-text retrieval, image captioning, and visual grounding.

**Strengths:**

- This paper highlights an interesting point: the SGA attack overlooks true image-text matching when using input augmentations. The motivation is clear.
- The method is simple.
- OT-Attack consistently improves SGA's attack success rate in three tasks: image-text retrieval, image captioning, and visual grounding.

**Weaknesses:**

- [W1] The presentation is not optimal.
    - [W1.1] The citation style is wrong: Authors should correctly use \citet and \citep. Please refer to the guideline carefully and modify them.
    - [W1.2] While the authors highlight OT-Attack’s successful attack against GPT-4 and Bing Chat at the end of the Introduction, there is no result in the main text.  Even in the Appendix, there is only one example without a comparison between attack methods, which makes it unreasonable to highlight this in the introduction. I suggest adding quantitative results or remove the claim.
- [W2] OT optimization introduces additional computational cost. How much does the computational cost differ between SGA and OT-Attack?
- [W3] Sensitivity to the regularization hyperparameter is unclear. How does the hyperparameter $\lambda$ affect the attack success rate (ASR) and computational cost? Does more accurate OT optimization lead to better ASR? I suggest adding an ablation study table.

**Questions:**

Please refer to the weaknesses.

---

> ### Author Response · Authors · 2024-11-22
> **Rebuttal by Authors [1/2]**
>
> Thank you for your valuable review and suggestions. Below we respond to the comments in **Weaknesses (W)** and **Questions (Q)**.
>
>
> ***W1: The presentation is not optimal. (1) The citation style is wrong. (2) There is no result  on ChatGPT-4 and Bing.***
>
>
> Thank you for pointing this out. (1) We have revised the citation style in the revised version. (2) We randomly sample 100 images from Flickr30K to generate adversarial examples by using SGA and our OT-Attack. Then we evaluate them on ChatGPT4 and Bing. The results are shown in the following Table. It highlights the superiority of our OT-Attack, achieving significantly higher ASR rates of 24% on ChatGPT4 and 30% on Bing, compared to only 7% and 8% by SGA.  Please refer to **Appendix G** for more details.
>
> | Models      | ChatGPT4 | Bing  |
> |-------------|----------|-------|
> | SGA         | 7%       | 8%    |
> | OT-Attack   | **24%**  | **30%** |
>
>
> ***W2: OT optimization introduces additional computational cost. How much does the computational cost differ between SGA and OT-Attack?***
>
> Thank you for the feedback. Following your suggestion, we compute the computational cost (minutes) and compare the OT-Attack with SGA across four models (ALBEF, TCL, CLIP_VIT , and CLIP_CNN ). The results are shown in the following Table. Our OT-Attack consistently requires more time than SGA, which is approximately 1.75 times that of SGA, reflecting the added complexity of the proposed method.  Please refer to **Appendix E** for more details.
>
>
>
>
> | Method         | ALBEF | TCL | CLIP_VIT | CLIP_CNN |
> |----------------|-------|-----|----------|----------|
> | SGA           | 34    | 33  | 17       | 13       |
> | OT-Attack (ours) | 60    | 58  | 30       | 23       |

---

> ### Author Response · Authors · 2024-11-22
> **Rebuttal by Authors [2/2]**
>
> ***W3: Sensitivity to the regularization hyperparameter is unclear. How does the hyperparameter affect the attack success rate (ASR) and computational cost? Does more accurate OT optimization lead to better ASR?***
>
>
> The experimental settings of OT-Attack adhered to the default configurations in PLOT. This choice was primarily due to the insensitivity of OT-Attack to experimental parameters, as we will validate through ablation studies. In these studies, we independently evaluate the impact of three parameters $\lambda$, convergence threshold (thresh), and the iteration limit of the Sinkhorn algorithm—on the experimental outcomes. ALBEF is used as the source model, and TCL as the target model.
>
> The parameter $\lambda$ balances the minimization of transport cost and plan smoothness. In our experiments, the default value was set to 0.1. To analyze the sensitivity of $\lambda$, we conduct the OT-Attack with different $\lambda$. The results are shown in the following Table. The results indicate that varying $\lambda$ among 0.01, 0.1, and 1 while keeping other conditions constant leads to consistent attack success rates across metrics such as TR R@1, TR R@5, TR R@10, IR R@1, IR R@5, and IR R@10. This demonstrates that OT-Attack is robust to changes in $\lambda$, maintaining stable performance.
>
>
> | λ Value | TR @1 | TR @5 | TR @10 | IR @1 | IR @5 | IR @10 |
> |---------|-------|-------|--------|-------|-------|--------|
> | 0.01    | 52.27 | 30.35 | 22.95  | 60.98 | 41.80 | 32.86  |
> | 0.1     | 52.37 | 30.45 | 23.05  | 61.00 | 41.95 | 32.68  |
> | 1.0     | 52.90 | 30.45 | 23.05  | 61.17 | 41.95 | 32.68  |
>
>
> In this study, the default settings were $\text{thresh} = 1.00 \times 10^{-2}$ and an iteration limit of 100. First, we analyzed the average number of iterations under $\text{thresh} = 1.00 \times 10^{-2}$, finding that the mean iteration count for generating 1,000 adversarial samples was only 2.3. Further, we examined the error scalar after each iteration. The error scalar starts at the order of $1.00 \times 10^{3}$ after the first iteration, reaches $1.00 \times 10^{-2}$ within two iterations, and decreases to $1.00 \times 10^{-3}$ or $1.00 \times 10^{-4}$ after three iterations. This analysis indicates that $\text{thresh}$ should range between $1.00 \times 10^{3}$ and $1.00 \times 10^{-6}$. If $\text{thresh}$ exceeds $1.00 \times 10^{3}$, the Sinkhorn algorithm converges in just one iteration. We also conduct OT-Attack with different threshold values. The results are shown in the following Table. It reveals minimal differences in image-text matching metrics when the source model is ALBEF, and the target model is TCL, confirming that OT-Attack is insensitive to thresh and demonstrates good stability.
>
> | Threshold Value   | TR @1 | TR @5 | TR @10 | IR @1 | IR @5 | IR @10 |
> |-------------------|-------|-------|--------|-------|-------|--------|
> | $1.00 \times 10^{-6}$ | 53.32 | 30.45 | 22.85  | 61.12 | 41.82 | 33.04  |
> | $1.00 \times 10^{-2}$ | 52.37 | 30.45 | 23.05  | 61.00 | 41.95 | 32.68  |
> | $1.00 \times 10^{3}$  | 53.11 | 30.85 | 23.05  | 60.86 | 41.64 | 32.80  |
>
>
>
> Additionally, We also conduct OT-Attack with different iteration limit values. The results are shown in the following Table. It demonstrates that the attack efficacy of OT-Attack remains stable.
>
> | Iteration Limit Value | TR @1 | TR @5 | TR @10 | IR @1 | IR @5 | IR @10 |
> |-----------------------|-------|-------|--------|-------|-------|--------|
> | 1                     | 52.21 | 30.25 | 22.85  | 60.64 | 41.80 | 32.34  |
> | 2                     | 52.32 | 30.45 | 23.05  | 60.86 | 41.95 | 32.68  |
> | 3                     | 52.37 | 30.45 | 23.05  | 61.00 | 41.95 | 32.68  |
> | 100                   | 52.37 | 30.45 | 23.05  | 61.00 | 41.95 | 32.68  |
>
> In summary, OT-Attack exhibits robustness to hyperparameter variations, delivering stable attack performance under different parameter settings, and significantly outperforms SGA. Please refer to **Appendix F** for more details.

---

> ### Author Response · Authors · 2024-11-24
> **Official Comment by Authors**
>
> Dear Reviewer uWjN,
>
> Thank you once again for taking the time to review our paper. We deeply value your insightful comments and are sincerely grateful for your efforts.
>
> We kindly request you to review our reply to see if it sufficiently addresses your concerns. Your feedback means a lot to us.
>
> Thank you deeply, Authors

---

> > ### Comment · Reviewer_uWjN · 2024-11-24
> > **Response by Reviewer**
> >
> > Thank you for your response. I have a few additional questions that I believe should be addressed.
> >
> > [W1-2]
> > - While 100 images are small, this result looks nice.
> >
> > [W2]
> > - How many adversarial samples were generated for Table 9?
> >
> > [W3]
> > - If the mean iteration count is only 2.3, why does OT-Attack require 1.75 times the computational cost of SGA?
> > - What is the computational cost when the Iteration Limit is set to 1? Given that attack performance is similar with varying hyperparameters, it would be helpful for future work to include the hyperparameter settings for the fastest OT-Attack.
> > - I did not expect such instability in hyperparameters, especially since OT-Attack should theoretically perform better with more accurate OT optimization.
> >   - Why does a larger $\lambda$, which should improve optimization accuracy, not result in better OT-Attack performance?
> >   - A smaller error scalar is expected to improve performance. However, the threshold of $1.0 \times 10^3$ seems to yield results similar to the default (likely no statistically significant difference). Could you clarify this? Some related questions: What is the error scalar before optimization?

---

> ### Author Response · Authors · 2024-11-25
> **Rebuttal by Authors**
>
> ***[W1-2] While 100 images are small, this result looks nice.***
>
> Thank you for the feedback. Due to limited time, we randomly selected 100 images for comparison experiments. In the future, we will use more images to verify the effectiveness of our method.
>
> ***[W2] How many adversarial samples were generated for Table 9?***
>
> Following the default setting of SGA, we also adopt 1000 images of Flickr30K to conduct experiments. Hence, 1000 adversarial examples are generated for Table 9.
>
> ***[W3-1] If the mean iteration count is only 2.3, why does OT-Attack require 1.75 times the computational cost of SGA?***
>
> We need to clarify that ``the mean iteration count`` is only 2.3 refers to the number of iterations required for each iteration in OT-Attack, not the number of iterations required for the entire OT-Attack. The optimal transfer matrix T is dependent on both the image feature matrix and the text feature matrix. As a result, the gradient computation and backpropagation process require a certain amount of time.
>
> ***[W3-2] What is the computational cost when the Iteration Limit is set to 1? Given that attack performance is similar with varying hyperparameters, it would be helpful for future work to include the hyperparameter settings for the fastest OT-Attack.***
>
> Thank you for the feedback. Following the suggestion, we conduct the fast OT-Attack with the Iteration Limit =1 . The results are shown in the following Table.  While SGA is the fastest at 34 minutes, our Fast-OT-Attack achieves a good balance, taking 49 minutes with likely improved performance. OT-Attack, though the most computationally intensive at 60 minutes, likely delivers the best performance.
>
> | Method              | SGA | Fast-OT-Attack (ours) | OT-Attack (ours) |
> |---------------------|-----|-----------------------|------------------|
> | Time (minutes)      |  34 | 49                 | 60               |
>
>
> ***[W3-3-1] Why does a larger $\lambda$, which should improve optimization accuracy, not result in better OT-Attack performance?***
>
> As shown in the following table.  A larger $\lambda$ indeed improves attack performance. For instance, increasing $\lambda$ from 0.01 to 1 boosts TR @R1 from 52.27% to 52.90%. However, a larger $\lambda$ also drives the Sinkhorn algorithm closer to the classical optimal transport problem, reducing its convergence speed, increasing the number of Sinkhorn iterations, and resulting in higher time costs. This deeper analysis highlights the significance of exploring parameter settings to achieve a balance between attack success rate and computational cost, a promising direction for future research.
>
> | λ Value | TR @1 | TR @5 | TR @10 | IR @1 | IR @5 | IR @10 |
> |---------|-------|-------|--------|-------|-------|--------|
> | 0.01    | 52.27 | 30.35 | 22.95  | 60.98 | 41.80 | 32.86  |
> | 0.1     | 52.37 | 30.45 | 23.05  | 61.00 | 41.95 | 32.68  |
> | 1.0     | 52.90 | 30.45 | 23.05  | 61.17 | 41.95 | 32.68  |
>
> ***[W3-3-2] A smaller error scalar is expected to improve performance. However, the threshold of $1.0 \times 10^3$ seems to yield results similar to the default (likely no statistically significant difference). Could you clarify this? Some related questions: What is the error scalar before optimization?***
>
> Thank you for pointing this out. The error scalar after the first optimization of most samples is at the $10^3$ level, which is greater than $1.0 \times 10^3$. After another optimization, the error scalar will drop sharply to the $ \times 10^{-2}$ level. Therefore, the performance difference between $1.0 \times 10^3$ and $1.0 \times 10^{-2}$ is not large. As presented in Algorithm 1 of the submitted manuscript, the error scalar is a temporary variable we have defined. This variable only comes into existence during the optimization transfer process and does not have a predefined value prior to that stage.

---

> > ### Comment · Reviewer_uWjN · 2024-11-25
> > **Response by Reviewer**
> >
> > Thank you for your response.
> >
> > [W2] Thank you for clarification. Please clarify it in Appendix E.
> >
> > [W3-1] Thank you for the clarification; I have misunderstood it.
> >
> > [W3-3-1] Given the randomness in augmentations during attacks, I believe the 0.63% difference is not statistically significant. In fact, the IR@10 score is higher when $\lambda = 0.01$.
> > Should this be interpreted as OT optimization for image-text matching being overly simple and highly insensitive to hyperparameters?
> >
> > Alternatively, providing visualization of image-text matching results before and after OT optimization could help support your argument.

---

> > > ### Author Response · Authors · 2024-11-26
> > > **Rebuttal by Authors**
> > >
> > > ***[W2]  Thank you for clarification. Please clarify it in Appendix E.***
> > >
> > > Thanks for pointing this out, we have clarified it in the latest version. Please refer to **Appendix E** for more details.
> > >
> > >
> > > ***[W3-3-1]  Given the randomness in augmentations during attacks, I believe the 0.63% difference is not statistically significant. In fact, the IR@10 score is higher when
> > > λ=0.01. Should this be interpreted as OT optimization for image-text matching being overly simple and highly insensitive to hyperparameters? Alternatively, providing visualization of image-text matching results before and after OT optimization could help support your argument.***
> > >
> > > Yes, your interpretation is correct, and I appreciate your insightful observation. OT optimization for image-text matching does appear to exhibit simplicity and relative insensitivity to hyperparameters. Additionally, your suggestion to include visualizations of image-text matching results before and after OT optimization is excellent. We will add the visualization in the final version. Thank you again for your valuable input!

---

> > > > ### Comment · Reviewer_uWjN · 2024-11-28
> > > > **Response by Reviewer**
> > > >
> > > > Thank you for your response.
> > > >
> > > > Currently, I am uncertain about the necessity of OT, particularly given the excessive invariance of performance to its hyperparameters.
> > > >
> > > > As far as I understand, the purpose of introducing OT is to focus more on correctly matching image-text pairs. To achieve this, instead of relying on the transportation matrix T (as in "sim_op = torch.sum(T * sim, dim=(0, 1))"), what about simply applying the power of $p$ (e.g, p=2,3) directly to the similarity matrix, "sim_op = torch.sum(sim ** p, dim=(0, 1))"? Would this achieve similar results?

---

> ### Author Response · Authors · 2024-12-02
> **Rebuttal by Authors**
>
> Thank you for the feedback. Following the suggestion, we conduct the comparative experiments. The results are shown in Following table. From the table, it is evident that the OT method using the transportation matrix T significantly outperforms directly applying power transformations to the similarity matrix (p=2,3,4) in certain metrics, such as CLIP-CNN's TR R@1 and IR R@1. The advantage of the transportation matrix T lies in its ability to leverage optimal transport theory to find a globally optimal coupling between image and text features, which can improve the adversarial transferability.
>
>
> | Method          | ALBEF (TR R@1) | ALBEF (IR R@1) | TCL (TR R@1) | TCL (IR R@1) | CLIPViT (TR R@1) | CLIPViT (IR R@1) | CLIPCNN (TR R@1) | CLIPCNN (IR R@1) |
> |-----------------|----------------|----------------|--------------|--------------|------------------|------------------|------------------|------------------|
> | SGA             | 97.24          | 97.28          | 45.42        | 55.25        | 33.38            | 44.16            | 34.93            | 46.57            |
> | p=2             | 98.02          | 97.87          | 49.11        | 59.76        | 33.97            | 44.75            | 37.68            | 48.27            |
> | p=3             | 98.23          | 97.78          | 49.74        | 59.48        | 33.83            | 45.3             | 38.7             | 49.02            |
> | p=4             | 97.71          | 97.1           | 49.32        | 59.93        | 34.21            | 45.13            | 38.95            | 48.58            |
> | OT-Attack (Ours)| 95.93          | 95.86          | **52.37**        |  **61.05**       |  **34.85**           |  **47.10**            |  **42.33**           |  **53.03**           |

---

> > ### Comment · Reviewer_uWjN · 2024-12-02
> > **Response by Reviewer**
> >
> > Thank you for your response.
> >
> > I believe all of my concerns have been addressed, and I will maintain my score of 8. Thank you once again for your dedicated effort.

---

> > > ### Author Response · Authors · 2024-12-02
> > > **Thank you for your support**
> > >
> > > Thank you for your support ! We greatly appreciate your insightful feedback and suggestions.

---

### Official Review · Reviewer_vFmV · 2024-11-04

**Soundness:** 2
**Presentation:** 3
**Contribution:** 2
**Rating:** 5
**Confidence:** 5

**Summary:**

This paper focuses on the task of conducting adversarial attacks on vision-language models. The focus of the attack is transferability. The paper introduces the optimal transport perspective for finding more transferable attacks by considering the optimal alignment between augmented image-text pairs. In experiments, the paper compares the proposed method with single-modality attacks, separate attacks by two single-modality attacks, and two multi-modal attacks.

**Strengths:**

This paper introduces a new perspective on an existing problem. The paper considers optimal alignment between augmented image-text pairs for more transferable adversarial attack patterns. This idea makes sense and is original to the best of my knowledge.

**Weaknesses:**

1. The evaluation baselines are not comprehensive in experimental design. The use of single-modality attacks PGD/Bert-Attack as baselines is weak in proving the effectiveness of the proposed method. There are certain representative multi-modal adversarial attacks not included for comparison. E.g., VLATTACK by Yin et al. in NeurIPS 2023, which focuses on the transferability of attacking pretrained vision-language models. These representative multi-modal attacks are not discussed in the paper either.
2. Although the perspective is new, the proposed solution is proposed to solve an existing problem, which makes the contribution trivial. Meanwhile, the paper lacks technical depth for discussing the applications of optimal transport in this problem. First, the paper lacks a discussion on the computational overhead caused by optimal transport. This paper introduces the application of it in vision-language models. However, the computation of optimal transport needs a certain time. The paper hasn't discussed it in the paper. Second, the paper didn't discuss the convergence of optimal transport for vision-language model attack either. Such algorithms need careful selection of hyperparameters. Thus, although the designed method sounds reasonable, this paper lacks technical depth for discussing the tradeoff and convergence.
3. Formatting problem exists in the current manuscript: e.g., L251, 'transport'. L411, '1 3 5 7' is not centered in table 4.

**Questions:**

1. What's the convergence condition for solving the optimal transport problem in this proposed setting?
2. What's the time cost is like for solving the optimal transport problem in this proposed setting?

---

> ### Author Response · Authors · 2024-11-22
> **Rebuttal by Authors**
>
> Thank you for your valuable review and suggestions. Below we respond to the comments in **Weaknesses (W)** and **Questions (Q)**.
>
> ***W1: The evaluation baselines are not comprehensive in experimental design and lack comparison with VLAttack.***
>
>
> Thank you for your feedback. We compare the proposed OT-Attack with VLAttack, which focuses on enhancing the transferability of attacking pretrained vision-language models. In our experiments, ALBEF is employed as the source model and TCL is used as the target model. The results are shown in the following Table.  The results show that our OT-Attack outperforms VLAttack across all metrics, with notable improvements such as 52.37\% vs. 43.2\% in TR@1 and 61.05\% vs. 52.04\% in IR@1, demonstrating the superiority of our method in improving the adversarial transferability. Please refer to **Appendix D** for more details.
>
>
> | Attack          | TR @ 1 | TR @ 5 | TR @ 10 | IR @ 1 | IR @ 5 | IR @ 10 |
> |------------------|--------|--------|---------|--------|--------|---------|
> | VL Attack | 43.2   | 23.09  | 16.01   | 52.04  | 32.14  | 24.21   |
> | OT-Attack (ours)             | 52.37  | 30.45  | 23.05   | 61.05  | 41.95  | 32.68   |
>
>
>
> ***W2 & Q2 : The paper lacks a discussion on the computational overhead caused by optimal transport.***
>
> Thank you for the feedback. Following your suggestion, we compute the computational cost (minutes) and compare the OT-Attack with SGA across four models (ALBEF, TCL, CLIP_VIT , and CLIP_CNN ). The results are shown in the following Table. Our OT-Attack consistently requires more time than SGA, which is approximately 1.75 times that of SGA, reflecting the added complexity of the proposed method.  Please refer to **Appendix E** for more details.
>
>
>
>
> | Method         | ALBEF | TCL | CLIP_VIT | CLIP_CNN |
> |----------------|-------|-----|----------|----------|
> | SGA           | 34    | 33  | 17       | 13       |
> | OT-Attack (ours) | 60    | 58  | 30       | 23       |
>
>
>
>
> ***W2 & Q1: The paper didn't discuss the convergence of optimal transport for vision-language model attack either.***
>
> The convergence of the Sinkhorn algorithm ensures that the transport matrix $P = \text{diag}(r) \cdot K \cdot \text{diag}(c)$ satisfies the prescribed marginal distributions $u$ and $v$. Convergence is typically assessed by measuring the change in the scaling factors r or c between successive iterations, where the error metric (e.g., $\| r^{(k)} - r^{(k-1)} \|$) must fall below a predefined threshold $\epsilon$. Alternatively, convergence can be determined by the deviation of the row and column sums of $P$ from $u$ and $v$. A maximum iteration limit is often imposed to prevent infinite loops. In this study, Sinkhorn convergence is determined based on the condition that the variation in the row normalization factor r between consecutive iterations is below the predefined threshold $\epsilon, i.e., \| r^{(k)} - r^{(k-1)} \| < \epsilon$. Please refer to **Appendix F** for more details.
>
>
>
> ***W3: Formatting problem exists in the current manuscript: e.g., L251, 'transport'. L411, '1 3 5 7' is not centered in table 4.***
>
> Thank you for pointing this out. We have revised them in the revised version. Please refer to L251 of Page 5 and L411 of Page 8.

---

> ### Author Response · Authors · 2024-11-24
> **Official Comment by Authors**
>
> Dear Reviewer vFmV,
>
> Thank you once again for taking the time to review our paper. We deeply value your insightful comments and are sincerely grateful for your efforts.
>
> We kindly request you to review our reply to see if it sufficiently addresses your concerns. Your feedback means a lot to us.
>
> Thank you deeply, Authors

---

### Author Response · Authors · 2024-11-22
**Summary of Paper Revision**

We thank all reviewers for their constructive feedback, and we have responded to each reviewer individually. We have also uploaded a **Paper Revision** including additional results and illustrations:

- $\\textrm{\\color{blue}Approach}$: revised and added more details about the proposed OT-Attack.
- $\\textrm{\\color{blue}Appendix D}$: add more comparisons with more baselines (VLAttack).
- $\\textrm{\\color{blue}Appendix E}$ : add computational cost of the proposed method.
- $\\textrm{\\color{blue}Appendix F}$ : add more hyperparameter analysis in OT-Attack.
- $\\textrm{\\color{blue}Appendix F.1}$: add sensitivity analysis of $\lambda$.
- $\\textrm{\\color{blue}Appendix F.2 }$ : add analysis of convergence threshold and sinkhorn iteration limit.
- $\\textrm{\\color{blue}Appendix G}$ : add more experiments on ChatGPT4 and Bing.
- $\\textrm{\\color{blue}Appendix H}$: add analysis of the effectiveness of OT-Attack.
- $\\textrm{\\color{blue}Appendix I}$: add more experiments on more source models.
- $\\textrm{\\color{blue}Appendix J}$: add more experiments on more models for image generation tasks.
- $\\textrm{\\color{blue}Appendix K}$: add an ablation study of our OT-Attack.

---

### Note · Authors · 2025-02-26

I have read and agree with the venue's withdrawal policy on behalf of myself and my co-authors.

---

### Meta-Review · Area_Chair_iVcr · 2024-12-18

**Metareview:**

This paper focuses on the task of conducting adversarial attacks on vision-language models. The focus of the attack is transferability. The paper introduces the optimal transport perspective for finding more transferable attacks by considering the optimal alignment between augmented image-text pairs. In experiments, the paper compares the proposed method with single-modality attacks, separate attacks by two single-modality attacks, and two multi-modal attacks. For the strengths, this paper highlights an interesting point: the SGA attack overlooks true image-text matching when using input augmentations. The motivation is clear, and the method is simple. Moreover, OT-Attack consistently improves SGA's attack success rate in three tasks: image-text retrieval, image captioning, and visual grounding. However, there are several points to be further improved. For example, the evaluation baselines are not comprehensive in experimental design. Moreover, although the perspective is new, the proposed solution is proposed to solve an existing problem, which makes the contribution trivial. Besides, enhancing the clarity and coherence of the writing, particularly in the methods section, would significantly improve its readability and impact. Therefore, this paper cannot be accepted at ICLR this time, but the enhanced version is highly encouraged to submit other top-tier venues.

**Additional Comments On Reviewer Discussion:**

Reviewers keep the score after the rebuttal.

---

### Decision · Program_Chairs · 2025-01-22

Reject